# Structural basis of broad protection against influenza virus by human antibodies targeting the neuraminidase active site via a recurring motif in CDR H3

Gyunghee Jo [1], Seiya Yamayoshi [2,3,4,5], Krystal M. Ma [6,7,8], Olivia Swanson [1], Jonathan L. Torres [1], James A. Ferguson[1], Monica L. Fernández-Quintero[1], Jiachen Huang[1], Jeffrey Copps [1], Alesandra J. Rodriguez[1], Jon M. Steichen[6,7,8], Yoshihiro Kawaoka [2,4,5,9], Julianna Han [1] ✉ & Andrew B. Ward [1] ✉

Influenza viruses evolve rapidly, driving seasonal epidemics and posing global pandemic threats. While neuraminidase (NA) has emerged as a vaccine target, shared molecular features of NA antibody responses are still not understood. Here, we describe cryo-electron microscopy structures of the broadly protective human antibody DA03E17, which was previously identified from an H1N1-infected donor, in complex with NA from A/H1N1, A/H3N2, and B/Victoria-lineage viruses. DA03E17 targets the highly conserved NA active site using its long CDR H3, which features a DR (Asp–Arg) motif that engages catalytic residues and mimics sialic acid interactions. We further demonstrate that this motif is conserved among several NA active site-targeting antibodies, indicating a common receptor mimicry strategy. We also identified BCR sequences containing this DR motif across all donors in a healthy human repertoire database, suggesting that such precursors may be relatively common and have vaccine targeting potential. Our findings reveal shared molecular features in NA active site-targeting antibodies that can be harnessed to design broad, immune-focused influenza vaccines.

Influenza viruses remain a significant global health threat, causing an estimated 290,000 to 650,000 seasonal influenza-associated respiratory deaths and 3 to 5 million cases of severe illness each year[1]. Currently, both influenza A viruses (IAVs; H1N1 and H3N2) and influenza B viruses (IBVs; Victoria-lineage) are co-circulating globally, while the B/ Yamagata-lineage has not been reliably detected since March 2020[2,3]. IAVs, in particular, undergo rapid antigenic evolution, leading to the emergence of new variants that may impact antibody recognition and pose new pandemic threats[4,5]. Recent antigenic drift in A/H3N2 viruses has introduced an N-glycosylation site at residue 245 on the surface

[1]Department of Integrative Structural and Computational Biology, The Scripps Research Institute, La Jolla, CA, USA. [2]Division of Virology, Institute of Medical Science, The University of Tokyo, Tokyo, Japan. [3]International Research Center for Infectious Diseases, Institute of Medical Science, University of Tokyo, Tokyo, Japan. [4]Research Center for Global Viral Infections, National Center for Global Health and Medicine, Tokyo, Japan. [5]The University of Tokyo Pandemic Preparedness, Infection and Advanced Research Center (UTOPIA), University of Tokyo, Tokyo, Japan. [6]Department of Immunology and Microbiology, The Scripps Research Institute, La Jolla, CA, USA. [7]IAVI Neutralizing Antibody Center, The Scripps Research Institute, La Jolla, CA, USA. [8]Center for HIV/AIDS Vaccine Immunology and Immunogen Discovery, The Scripps Research Institute, La Jolla, CA, USA. [9]Department of Pathobiological Sciences, School of Veterinary Medicine, University of Wisconsin-Madison, Madison, WI, USA. ✉e-mail: juliannahan@scripps.edu; andrew@scripps.edu

glycoprotein neuraminidase (NA) since the 2014/2015 season, which can shield certain epitopes and is known to reduce the activity of monoclonal and human serum NA-specific antibodies[6–8]. In addition, a highly pathogenic avian influenza (HPAI) H5N1 strain is currently spreading among cattle and other mammals in the United States[9,10]. This unprecedented outbreak has resulted in virus spillover into humans, raising concerns about zoonotic transmission and potential pandemic risks[11,12].

NA is one of the two surface glycoproteins of influenza viruses and facilitates the release of progeny viruses by cleaving the α2,3- or α2,6-linkage between sialic acid and the underlying galactose residue on host glycoproteins[13]. The NA active site is highly conserved due to functional constraints related to sialic acid recognition[14]. Hemagglutinin (HA), another surface glycoprotein, mediates viral entry by binding to sialic acid displayed on host cells[15]. Due to the essential role of NA in the viral life cycle, its catalytic activity has been the target of small-molecule drugs such as oseltamivir[16], although resistance-associated mutations such as H274Y have been reported[17,18]. Unlike HA, antigenic drift in NA occurs at a slower rate and independently[19,20]. Moreover, antibodies targeting NA are an independent correlate of protection with broader cross-reactivity[21,22], making NA a promising vaccine target for inducing protective antibodies less sensitive to seasonal drift[23]. However, current influenza vaccines primarily induce antibody responses against HA, with little or no targeting of NA[24], which can confer protection or reduce disease severity but often fail to prevent infection due to rapid antigenic drift in HA, necessitating annual updates[25,26].

Identifying recurring molecular features of broadly protective antibodies across individuals is essential for designing effective immunogens in the development of universal influenza vaccines[27,28]. The highly conserved HA stem has frequently been targeted by multidonor class antibodies with shared sequence features[29–33], and structural insights from these antibodies have been instrumental in advancing HA-based universal vaccine efforts[34,35]. In recent years, several monoclonal antibodies (mAbs) targeting the conserved NA active site have been identified[36–39]. These antibodies exhibit broader cross-reactivity across diverse influenza strains than other NA-specific antibodies, offering important insights for NA-based vaccine design. Broadly cross-reactive anti-NA mAb 1G01, isolated from an H3N2-infected individual, inhibits NA activity with exceptional breadth by blocking the NA active site with a long complementarity-determining region (CDR) H3 loop, engaging multiple conserved residues that also interact with sialic acid[36]. A broadly reactive anti-IBV NA mAb 1G05 also targets the active site using CDR H3, where residues D100a and R100b interact with conserved residues, with D100a mimicking the carboxylate group of sialic acid[37]. FNI9, identified more recently, similarly engages the NA active site via residues R106 and D107 in its CDR H3, mimicking key interactions of the sialic acid and achieving unprecedented breadth in inhibition across diverse influenza strains[39].

Nevertheless, our understanding of broadly protective NA epitopes and recurring molecular features of NA antibody responses remains limited compared to HA, underscoring the need to identify more broadly protective antibodies[28,40]. This would enable clearer patterns of molecular convergence to emerge, ultimately informing the design of effective universal influenza vaccines.

We previously identified and functionally characterized the human mAb DA03E17, which was isolated from an individual infected with the A/H1N1pdm09 virus during the 2015–2016 influenza season[38]. DA03E17 exhibited broad cross-reactivity against NAs from both group 1 and group 2 IAVs, as well as both lineages of IBVs. DA03E17 bound to or near the enzymatic active site of NA, effectively inhibiting sialidase activity, neutralizing diverse influenza strains in vitro, and providing protection in vivo across multiple subtypes of IAVs and IBVs. However, the precise epitope targeted by DA03E17 was not defined.

In this study, we present the cryo-electron microscopy (cryo-EM) structures of DA03E17 in complex with NA from A/H1N1, A/H3N2, and B/Victoria-lineage viruses. DA03E17 retained binding to NAs with oseltamivir-resistant mutations and showed strong affinity for NAs from recent H3N2 strains and the Bovine HPAI H5N1 virus, demonstrating its potential to target both seasonal and emerging zoonotic viruses. Structural analysis revealed that DA03E17 engages the highly conserved NA active site using its long CDR H3, where the DR (Asp–Arg) motif mimics the sialic acid receptor by interacting with key catalytic residues. This sialic acid mimicry mediated by the DR motif was structurally conserved across multiple NA active site-targeting antibodies, and deep repertoire analysis demonstrated that precursors to these antibodies are present in the human naive B cell pool, supporting the feasibility of targeting them through vaccination. In addition, we identified and structurally characterized additional DR motif antibodies from patents and the literature, providing further evidence for structural and mechanistic convergence within this antibody class. Together, these findings reveal a convergent mechanism of receptor mimicry by NA active site-targeting antibodies and underscore their potential as targets for NA-based universal influenza vaccines.

## Results
### Binding properties of DA03E17 to antiviral-resistant variants and diverse influenza NAs
We previously isolated and characterized a human mAb DA03E17 from peripheral blood mononuclear cells (PBMCs) of an individual who was infected with A/H1N1pdm09 virus in the 2015–2016 influenza season[38]. DA03E17 showed broad cross-reactivity against NAs from IAV group 1 (N1, N4, N5, N8), group 2 (N2, N3, N6, N7, N9), as well as both lineages of IBV (B/Yamagata and B/Victoria) (Supplementary Fig. 1a)[38]. Furthermore, DA03E17 inhibited the sialidase activity of NA, neutralized both IAVs and IBVs in vitro, and provided in vivo protection against several subtypes of influenza virus[38]. DA03E17 uses the *IGHV4-31* and *IGKV1-12* heavy and light chain V genes, respectively, and has a long CDR H3 (19 amino acids in the IMGT CDR definition scheme) (Supplementary Fig. 2). In addition to the previously reported broad reactivity of DA03E17, here we further characterized its ability to retain binding to NAs containing oseltamivir-resistant mutations. We generated recombinant N1 NA from H1N1 A/Brisbane/02/2018 (BB18), which contains previously reported stabilizing mutations derived from a computationally-designed NA (stabilized NA protein, sNAp) in the inter-protomeric interface to maintain a closed tetrameric state[41]. We produced BB18 N1 sNAp containing either the major oseltamivir-resistance mutation H274Y[17] or other resistance substitutions, I222V or S246N (N2 numbering)[18]. In addition, we generated recombinant N2 NA from H3N2 A/Indiana/10/2011 (IN11) with either the E119V or I222L substitution[42]. We measured the binding of DA03E17 to these recombinant N1 and N2 NAs by ELISA. While DA03E17 retained its binding to BB18 N1 sNAp with either the H274Y, I222V, or S246N substitution, it exhibited reduced binding to IN11 N2 with either the E119V or I222L substitution, although some level of binding was still observed (Supplementary Fig. 1b). Compared to DA03E17, the previously described broadly neutralizing anti-NA mAb 1G01[36] showed a greater difference in binding to oseltamivir-resistant BB18 N1 NAs and was particularly affected by the H274Y substitution, consistent with a previous report[39], while its binding to oseltamivir-resistant IN11 N2 NAs remained relatively consistent (Supplementary Fig. 1b). We also assessed the NI activity of DA03E17 using ELLA with the same recombinant NA proteins containing oseltamivir-resistant mutations. DA03E17 retained inhibitory activity against BB18 N1 NAs carrying these mutations, with enhanced inhibition observed for H274Y, while activity was reduced against S246N and especially I222V variants. In contrast, 1G01 showed reduced inhibition against S246N and H274Y variants, consistent with its ELISA binding profile. Oseltamivir-resistant IN11 N2 mutants were excluded from ELLA due to a lack of detectable sialidase activity, but DA03E17 exhibited similar levels of inhibition to 1G01 against the wild-type IN11 N2 (Supplementary Fig. 1c). These results

demonstrate that DA03E17 retains binding and inhibitory activity against several oseltamivir-resistant NA variants, while also revealing mutation-specific differences in sensitivity.

Biolayer interferometry (BLI) indicated that DA03E17 IgG bound to recombinant NAs from H1N1 A/California/07/2009 (CA09 N1 sNAp), H3N2 A/Perth/16/2009 (PT09 N2), and H3N2 A/Indiana/08/2011 (IN11 N2) with sub-picomolar apparent affinities, while it bound to NAs from B/Colorado/06/2017 (B/Victoria-lineage; CO17 B) and more recent H3N2 strains, including A/Kansas/14/2017 (KS17 N2) and A/Hong Kong/2671/2019 (HK19 N2), with sub-nanomolar to weaker nanomolar apparent affinities (Supplementary Fig. 3). Notably, DA03E17 IgG also bound to recombinant N1 sNAp derived from the HPAI H5N1 clade 2.3.4.4b virus (A/dairy cattle/Texas/24-008749-001/2024; TX24) with nanomolar apparent affinity, which is currently spreading across dairy herds and other mammals in 14 states in the United States[9–11] (Supplementary Fig. 3). DA03E17 Fab exhibited similar subtype-dependent binding patterns, with nanomolar to weaker nanomolar affinities for CA09 N1 sNAp, TX24 N1 sNAp, PT09 N2, and IN11 N2, and sub-micromolar affinities for KS17 N2, HK19 N2, and CO17 B. The binding of DA03E17 Fab to CO17 B NA was markedly reduced compared to IgG, indicating a greater contribution of avidity for this lineage (Supplementary Fig. 3). Together, these results confirm the broad cross-reactivity of DA03E17 to diverse influenza A and B virus NAs.

## Cryo-EM structures of DA03E17 Fab in complex with N1, N2, and B NAs

To elucidate the epitope of DA03E17 and the structural basis for its broad cross-reactivity, we determined the cryo-EM structures of the DA03E17 Fab in complex with CA09 N1 sNAp, and KS17 N2 and CO17 B NAs at 2.67, 2.86, and 2.47 Å resolution, respectively (Supplementary Fig. 4 and Supplementary Table 1). In addition, a cryo-EM structure of KS17 N2 NA was determined in its apo-form at 2.75 Å resolution using the same data set of DA03E17-KS17 N2 NA complex (Supplementary Fig. 4). In all three complex structures, DA03E17 binds in a similar orientation to all three NAs, with each Fab interacting with just one protomer of the NA tetramer (Fig. 1a–c and Supplementary Fig. 5a–f). DA03E17 fully blocks the NA active site by protruding the CDR H3 into the active site pocket (Fig. 1c and Supplementary Fig. 5c, f), consistent with our previous study using the small molecule-based NA-Star assay, which demonstrated that the NA inhibition (NI) activity of DA03E17 is due to direct inhibition[38]. No large conformational changes in the global structure of the NA protein are observed compared with corresponding wild-type structures, as indicated by an RMSD of 0.356, 0.402, and 0.314 Å across all pairs for the N1, N2, and B NAs, respectively (Supplementary Fig. 5g–i).

In the DA03E17-CA09 N1 sNAp complex structure, the buried surface area (BSA) of the DA03E17 and N1 NA interface is 1022.2 Å², with the heavy chain accounting for 78% of the interaction. DA03E17 interacts with CA09 N1 sNAp using all CDR loops except CDR L2 (Fig. 1d), and the DA03E17 epitope on CA09 N1 sNAp consists of 32 residues around the active site pocket (Fig. 1e). The 19-residue CDR H3 contributes most of the Fab interactions with 19 N1 NA residues including seven catalytic site residues (R118, D151, R152, R224, R292, R371, and Y406) and six framework residues (E119, W178, I222, E227, E277, and N294) (Fig. 1f) which are strictly conserved across influenza A group 1, group 2, and influenza B NAs (Supplementary Table 2). Substitutions at D151 (D151G and D151N), which were previously identified as major escape mutations for DA03E17 in our earlier study[38], can be explained by the direct contact between the CDR H3 of DA03E17 and D151 observed in our structure (Fig. 1f). In contrast, although a substitution at T439 (T439A) was also identified as an escape mutation, no direct contact between DA03E17 and T439 was observed in our structures. This substitution may affect binding indirectly by altering the conformation of the NA active site. The CDRs H1, H2, L1, L3, and framework region (FR) L3 predominantly interact with residues

located at the periphery of the active site (I149, K150, D198, N199, N247, W295, H296, S342, N344, G345, A346, N347, R430, P431, and K432), further strengthening binding (Fig. 1g, h). Overall, DA03E17 utilizes both heavy and light chain CDRs to bind the active site of NA, and a total of 24 hydrogen bonds were observed between DA03E17 and CA09 N1 sNAp. In the DA03E17-KS17 N2 and DA03E17-CO17 B NA complex structures, the BSAs at the interfaces are 1224.5 Å² and 977.6 Å², respectively, indicating a slightly larger interaction surface with the KS17 N2 NA compared to the CA09 N1 and CO17 B NAs. Most of the interactions mediated by the CDR H3 in the DA03E17-KS17 N2 and DA03E17-CO17 B NA complexes are similar to those observed in the CA09 N1 sNAp complex (Supplementary Fig. 6a, e), with key epitope residues being conserved across IAV and IBV NAs (Supplementary Table 2), maintaining a consistent binding mode. These data establish that DA03E17 directly targets the highly conserved residues in the enzymatic active site of NA using its long CDR H3, thereby blocking the active site and inhibiting sialidase activity of NA.

## DA03E17 targets highly conserved epitopes in N1, N2, and B NAs

To further understand the broad cross-reactivity of DA03E17, we examined the sequence conservation of DA03E17 epitopes on NAs from H1N1, H3N2, and IBV viruses that have been circulating for several decades. (Fig. 2a–c). The epitopes of DA03E17 are well conserved among human seasonal H1N1 (circulating from 1977 to 2023) and H3N2 (from 1968 to 2023) IAVs, as well as B/Victoria/2/87-like IBVs (from 1987 to 2023). The majority of epitope residues are highly conserved or conservatively substituted on H1N1 (25 out of 32, 78%), H3N2 (25 out of 35, 71%), and B/Vic (25 out of 30, 83%) NAs (Fig. 2d). Notably, although the B/Yamagata-like lineage has been considered extinct since 2020[2,3], the DA03E17 B/Vic NA epitope is also highly conserved in NAs of B/Yamagata/16/88-like IBVs (from 1988 to 2020; 25 out of 30, 83%). We also assessed epitope conservation in animal-origin IAVs, analyzing N1 and N2 NA sequences from HxN1 and HxN2 viruses that have circulated in avian and mammalian hosts between 1977 and 2023 (Supplementary Fig. 7). DA03E17 epitope residues on N1 were generally well conserved in avian and mammalian HxN1 NAs, similar to human seasonal H1N1 viruses. In contrast, conservation in HxN2 NAs from animal-origin IAVs was lower compared to human H3N2 NAs. Substitutions at positions 199 and 221—the two most variable residues within the N1 and N2 epitopes—from N199 (CA09 N1) to S199 (H1N1 A/Yokohama/94/2014) and N221 (CA09 N1) to K221 (H1N1 A/Puerto Rico/8/1934) in N1, as well as from K199/D221 (KS17 N2) to E199/K221 (H3N2 A/Fujian/411/2002; FJ02) in N2, did not compromise the NI activity of DA03E17, as shown in our previous study[38]. Similarly, substitutions in the N2 epitope, from N147/R150/K344 (KS17 N2) to D147/H150/E344 (FJ02 N2), did not impair the NI activity of DA03E17, and substitutions in the influenza B NA epitope, from S244/R295/K343/E436 (CO17; B/Yamagata-lineage) to P244/S295/E343/T436 (B/Phuket/3073/2013; B/Victoria-lineage), had almost no effect on its NI activity[38]. These results suggest that DA03E17 binding is resilient to these substitutions. The core of the DA03E17 epitope in the active site, which is targeted by the CDR H3, is strictly conserved across A/H1N1, A/H3N2, and B/Victoria-like viruses due to the functional constraints related to sialic acid receptor recognition (Fig. 2d, e). Therefore, the broad cross-reactivity of DA03E17 can be partially explained by the strict conservation or conservative substitution of key epitope residues, along with its resilience to substitutions around the NA active site. Notably, several conserved residues in the active site pocket are charged amino acids, forming distinct charged patches with R118, R292, and R371 contributing to positive charges, and E119, D151, E227, and E277 forming negative charges (Fig. 2e, f). D100b and R100c, which form the DR (Asp–Arg) motif at the tip of the DA03E17 CDR H3, have an electrostatic surface that is complementary to these charged patches, enabling extensive salt bridge formation for targeted interaction (Figs. 1f, 2g, and Supplementary Fig. 6a, e). Overall, we found that the

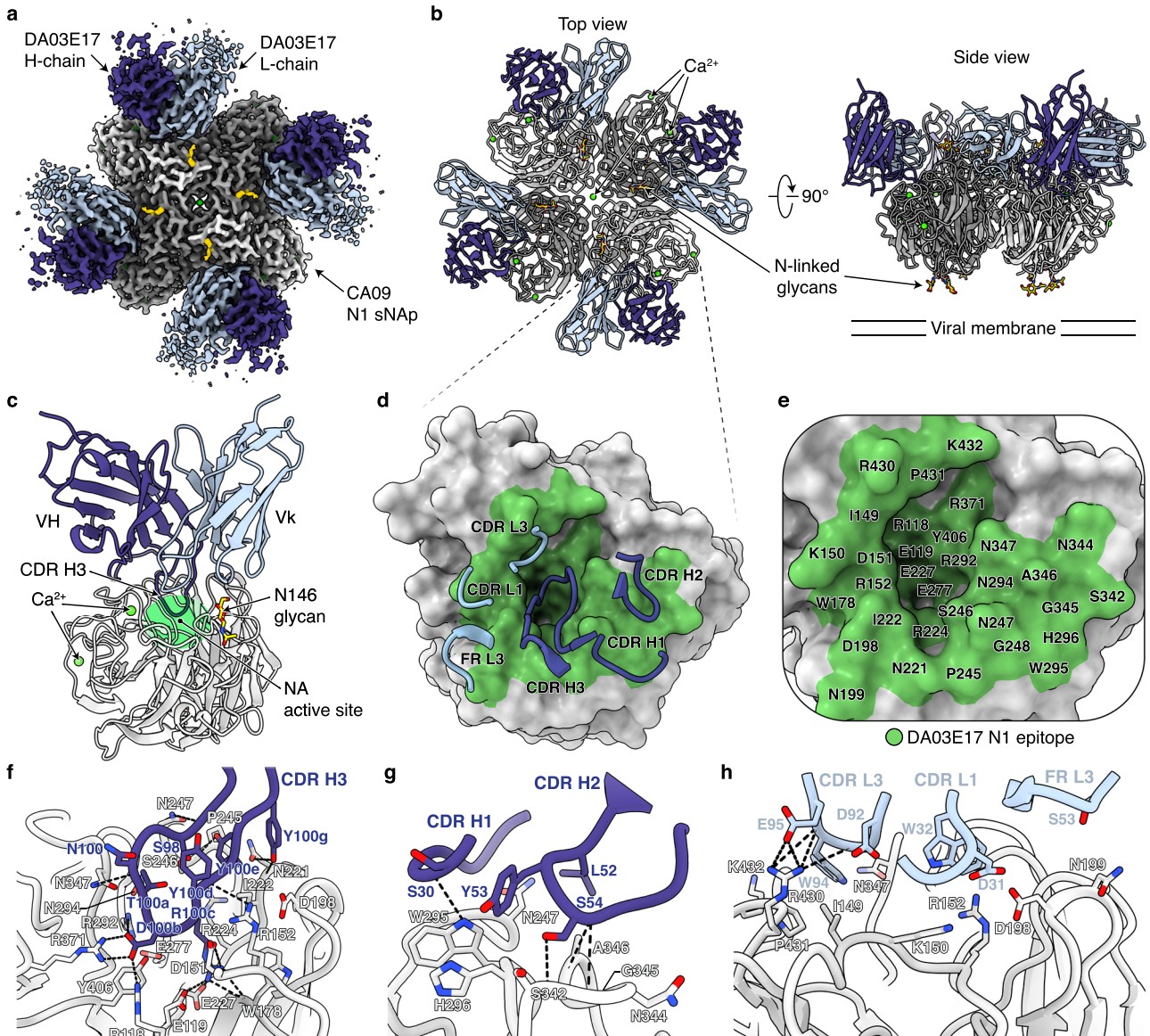

**Fig. 1 | Cryo-EM structure of DA03E17 Fab in complex with CA09 N1 sNAp.** **a**, **b** Cryo-EM map at 2.67 Å resolution (**a**) and two orthogonal views of the atomic model from the top and side (**b**); only the Fab variable region is built into the density. DA03E17 heavy chain, dark blue; light chain, light blue; NA head domain tetramer; shades of gray; NA glycans, gold; Ca²⁺ ion, lime. **c** Ribbon diagram of the CA09 N1 sNAp protomer bound to one DA03E17 Fab. The NA active site targeted by CDR H3 is highlighted by a green oval. VH, heavy chain variable domain; Vk, kappa light chain variable domain. **d**, **e** DA03E17 epitope mapped on the NA protomer surface with CDR loops involved (**d**) or residue labels (**e**). The DA03E17 N1 epitope is highlighted in green. **f**–**h**, Detailed interactions between CA09 N1 sNAp and DA03E17 CDR H3 (**f**), CDR H1 and H2 (**g**), and CDR L1, L3, and FR L3 (**h**).

DA03E17 epitope is largely conserved, especially the active site residues targeted by the CDR H3, which are strictly conserved across the NAs of H1N1, H3N2, and B/Victoria-like viruses that have been circulating for several decades, highlighting the critical role of the CDR H3 in the broad cross-reactivity of DA03E17.

### DA03E17 accommodates the N-glycans of the recently circulating human H3N2 viruses

The NA of circulating human A/H3N2 viruses has undergone substantial antigenic drift since 2014, resulting from mutations at positions 245 and 247 that introduced the N245 glycan near the active site[6] (Fig. 3a and Supplementary Fig. 8a). By the 2016/17 season, nearly all circulating H3N2 viruses carried the N245 glycan, which has become fixed in recent strains and has been shown to shield the NA active site, reducing the efficacy of certain NA active site-targeting antibodies[6–8,39]. Another N-linked glycan near the NA active site, the N146 glycan

(Fig. 3a), is highly conserved in human and animal IAVs, including human H1N1 viruses circulating since 1918 and human H3N2 viruses circulating since 1968[43], and a corresponding N144 glycan is also conserved in IBVs. Our previous study showed that DA03E17 maintained inhibitory activity against the NAs of recently circulating H3N2 viruses, despite the presence of the N245 glycan[38]. BLI analysis revealed that while DA03E17 Fab exhibited reduced affinities for drifted N2 NAs (KS17 and HK19) carrying the N245 glycan, as well as mutant IN11 N2 NA (IN11 N2 NAT) with an introduced N245 glycosylation site, compared to NAs without the N245 glycan, it still maintained submicromolar affinities (Fig. 3b and Supplementary Fig. 3). These results indicate that the N245 glycan may affect the binding of DA03E17 to the NA active site. To investigate in detail how the N245 glycan affects the binding of DA03E17, we compared the structure of the DA03E17-KS17 N2 NA complex with that of the apo-KS17 N2 NA, which was determined from the same data set of the DA03E17-KS17 N2 NA complex

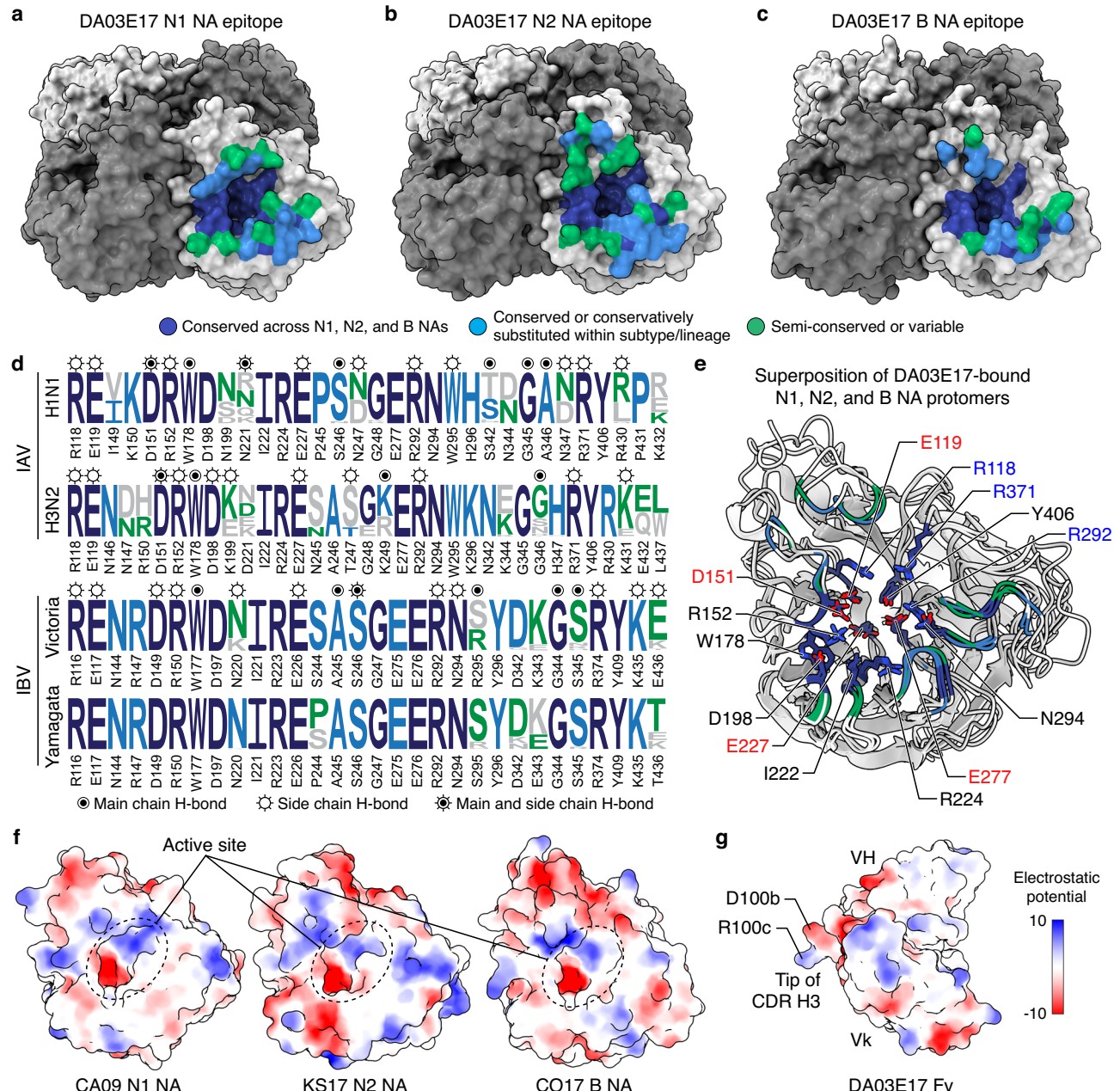

**Fig. 2 | Sequence conservation of the DA03E17 epitope. a–c** Sequence conservation of the DA03E17 epitope mapped onto surface representations of CA09 N1 (**a**), KS17 N2 (**b**), and CO17 B (**c**) NAs. NA residues conserved across A/H1N1, A/H3N2, and B/Victoria-like viruses are colored in dark blue, while residues conserved or conservatively substituted within each subtype or lineage are shown in sky blue. Semi-conserved or variable residues are colored in green. **d** Conservation of DA03E17 epitope residues based on NA sequences from human seasonal H1N1 (1977–2023) and H3N2 (1968–2023) IAVs, B/Victoria-like (1987–2023), and B/Yamagata-like (1988–2020) IBVs. Symbols mark NA residues forming hydrogen bonds with DA03E17: circled bullets for main chain, open circles with rays for side chain, and circled bullets with rays for both. **e** Superimposed protomers of CA09 N1 sNAp, and KS17 N2 and CO17 B NAs with conserved active site residues shown as sticks and key charged residues highlighted. **f, g** Surface representations of CA09 N1, KS17 N2, and CO17 B NA protomers (**f**) and DA03E17 Fv (**g**) colored by electrostatic potential from −10 to +10 $k_BT/e_c$. Dotted circles indicate charged patches in the NA active site pocket.

(Supplementary Fig. 4). In the apo-KS17 N2 structure, the N245 glycan would cause a steric clash with DA03E17 CDR H1, while the N146 glycan would clash with CDR L1. However, upon binding to KS17 N2 NA, DA03E17 induces conformational changes in both the N245 and N146 glycans, allowing it to maintain binding by repositioning these glycans and avoiding the clashes (Fig. 3c and Supplementary Fig. 8b).

To further investigate the structural changes in the N245 and N146 glycans upon DA03E17 binding to KS17 N2 NA, we performed molecular dynamics (MD) simulations. In the apo-KS17 N2 NA structure, the MD simulations revealed that the N245 and N146 glycans formed clusters that

obstructed access to the NA active site and potentially blocked the binding of DA03E17 (Supplementary Fig. 8c, d). In contrast, MD simulations of the DA03E17-KS17 N2 NA complex demonstrated that DA03E17 binding induced significant structural changes in both the N245 and N146 glycans. The glycans shifted to adopt specific conformations, moving away from the DA03E17 binding interface. These movements aligned with the cryo-EM structural data (Fig. 3a, c), further supporting that DA03E17 induces a rearrangement of the N245 and N146 glycans to facilitate stable binding to the NA active site. Such conformational changes in the glycans may require additional energy, potentially contributing to the reduced

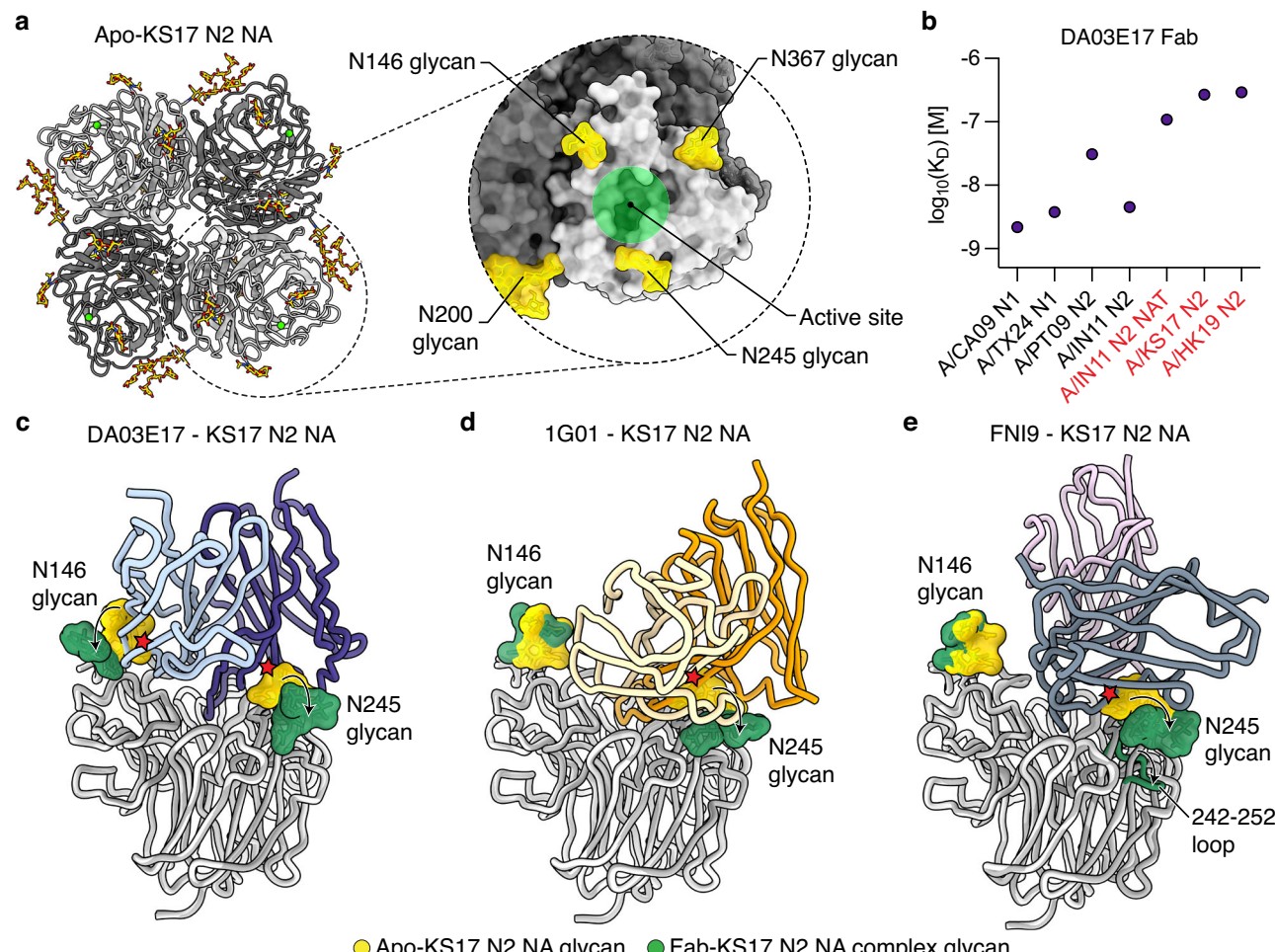

**Fig. 3 | DA03E17 accommodates the N-glycans of drifted N2 NA. a** Cryo-EM structure of KS17 N2 NA in apo-form at 2.75 Å resolution. The inset shows a magnified view of a KS17 N2 NA protomer, with the active site indicated by a green circle, and the surrounding glycans are shown in yellow. **b** Binding affinity of DA03E17 Fab to recombinant NAs, measured by BLI with CA09 N1, TX24 N1, PT09 N2, IN11 N2, IN11 N2 NAT mutant, KS17 N2, and HK19 N2 NAs. NAs with an N-glycan at position 245 are highlighted in red. $K_D$ was estimated using a 1:1 binding model.

Source data are provided as a Source Data file. **c–e** Overlay of the apo-KS17 N2 NA protomer (gray) with Fab-bound KS17 N2 NA protomers (white): DA03E17 Fab (**c**), 1G01 Fab (**d**), and FNI9 Fab (**e**). The N146 and N245 glycans on the apo-KS17 N2 NA are shown in yellow, and those on the Fab-bound KS17 N2 NA are shown in green. Steric clash shown with red stars. Conformational changes in glycans and the 242–252 loop are indicated by arrows.

affinity of DA03E17 for N2 NAs carrying the N245 glycan. Considering that the N146 glycan is also present in other N1 and N2 NAs, such as CA09 N1, PT09 N2, and IN11 N2, where DA03E17 Fab binds with a range of nanomolar affinity (Supplementary Fig. 3), the N245 glycan, either alone or in the context of the N146 glycan, plays a prominent role in the observed reduction in apparent affinity for recent N2s.

Previous studies have also explored how NA active site-targeting antibodies maintain binding and protective efficacy against contemporary H3N2 viruses carrying the N245 glycan, offering insights into the mechanisms these antibodies employ to accommodate this glycan[8,39]. The broadly cross-reactive antibody, 1G01, was shown to maintain protective efficacy in vivo despite reduced inhibition against viruses harboring the N245 glycan[8]. Although structural analysis using negative-stain electron microscopy indicated that 1G01 still binds the active site of N2 NA with the N245 glycan, the precise mechanism remained unclear. Another broadly cross-reactive antibody, FNI9, was shown to not only induce conformational changes in the N245 glycan but also alter the conformation of the 242–252 loop of HK19 N2 NA, which contains the $N_{245}AT_{247}$ glycosylation motif[39]. To compare the N245 glycan accommodation mechanisms of 1G01 and FNI9 with that of DA03E17 directly, we determined the cryo-EM structures of the

1G01-KS17 N2 NA and FNI9-KS17 N2 NA complexes at 3.20 Å and 2.97 Å resolutions, respectively (Supplementary Fig. 9). Structural comparison revealed that CDR H3 of 1G01 would clash with the N245 glycan in the apo-KS17 N2 NA structure. However, similar to DA03E17, 1G01 induces conformational changes in the N245 glycan, shifting it away from the binding interface and maintaining binding to the NA (Fig. 3d). For FNI9, similar to previous findings with HK19 N2 NA[39], structural comparison revealed that it induces conformational changes not only in the N245 glycan but also in the 242–252 loop of KS17 N2 NA (Fig. 3e). This rearrangement allows FNI9 to maintain binding, suggesting that its glycan accommodation mechanism is not HK19 N2 NA-specific but rather a broader adaptation to drifted N2 NAs carrying the N245 glycan in general. To summarize, the three broadly cross-reactive antibodies, DA03E17, 1G01, and FNI9, accommodate the N245 glycan on drifted N2 NA through two primary mechanisms. DA03E17 and 1G01 induce conformational changes in the glycan, shifting it away from the binding interface, while FNI9 not only induces glycan rearrangement but also alters the conformation of the 242–252 loop, indicating a more extensive structural rearrangement. These results demonstrate two distinct strategies employed by NA active site-targeting antibodies to maintain effective active site binding.

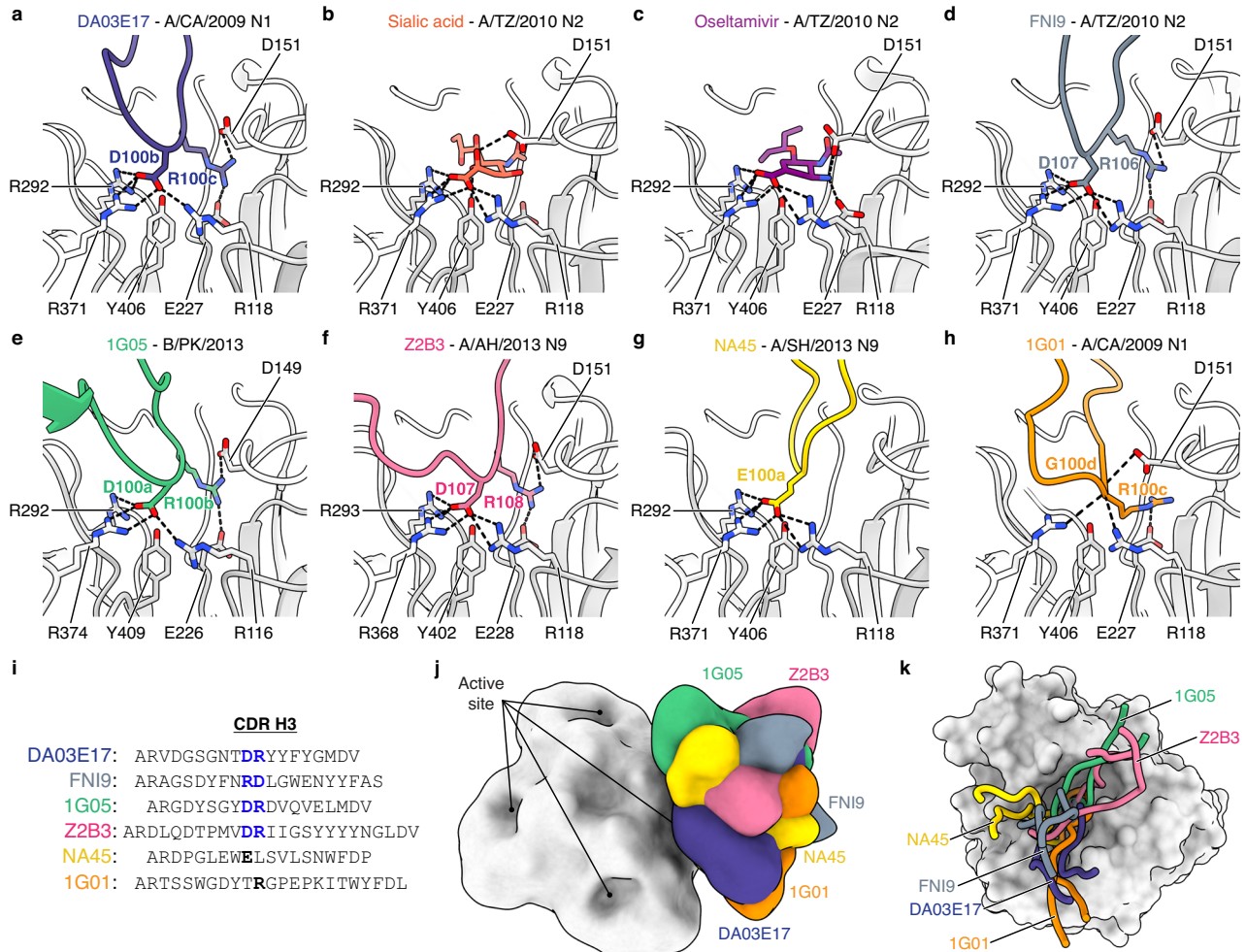

**Fig. 4 | The DR motif is a recurring molecular feature in the CDR H3 of NA active site-targeting antibodies. a** Network of salt bridge interactions between D100b and R100c in the CDR H3 of DA03E17 and active site residues of CA09 N1 sNAp. **b, c** Salt bridge interactions between sialic acid (**b**) and oseltamivir (**c**) with active site residues of H3N2 A/Tanzania/205/2010 (TZ10) NA (PDB: 4GZQ, 4GZP). **d** Salt bridge interactions between D107 and R106 in the CDR H3 of FNI9 and TZ10 N2 NA (PDB: 8G3N). **e** Salt bridge interactions between D100a and R100b in the CDR H3 of 1G05 and B/Phuket/3073/2013 NA (PDB: 6V4N). **f** Salt bridge interactions between D107 and R108 in the CDR H3 of Z2B3 and H7N9 A/Anhui/1/2013 NA (PDB: 6LXJ).

**g** Salt bridge interactions between E100a in the CDR H3 of NA45 and H7N9 A/Shanghai/2/2013 NA (PDB: 6PZE). **h** Hydrogen bonds and salt bridge interactions between R100c in the CDR H3 of 1G01 and CA09 N1 NA (PDB: 6Q23). **i** Alignment of the CDR H3 sequences of DA03E17, FNI9, 1G05, Z2B3, NA45, and 1G01, with DR motifs mimicking the interaction of sialic acid highlighted in red. **j, k** Overlay of low-pass filtered structures of NA active site-targeting antibodies bound to the tetrameric NA head domain (**j**), and a detailed view of their CDR H3 loops occupying the active site pocket of the NA protomer (**k**).

## DA03E17 mimics the interaction of sialic acid using a conserved motif in the CDR H3

The DA03E17 footprint on the NA spans the entire active site pocket, fully covering it and blocking access to critical residues involved in enzymatic activity (Fig. 1e), suggesting that DA03E17 may inhibit NA activity through a mechanism similar to known NA inhibitors, which block the active site by mimicking sialic acid interactions[44,45]. We compared the interaction formed between the DA03E17 CDR H3 and the NA active site residues with that of the sialic acid receptor and oseltamivir. Remarkably, the carboxylate side chain of D100b in the DA03E17 CDR H3 forms the same salt bridge interaction network with NA residues R118, R292, and R371 as the carboxylate group of sialic acid and oseltamivir (Fig. 4a–c). The side chain of R100c in the CDR H3 also forms contacts with NA residues D151 and E227. While sialic acid and oseltamivir also contact these residues, R100c in the CDRH3 contacts through different interactions, forming a salt bridge that strengthens binding. Such receptor mimicry by antibody CDR H3 has been previously observed in other NA active site-targeting antibodies[37,39,46]. In our structural comparison, we found that the DR (Asp–Arg) motif in DA03E17 functions similarly to those in the previously reported

pan-influenza NA mAb FNI9[39] and the broadly cross-reactive anti-IBV NA mAb 1G05[37], mimicking sialic acid interactions and blocking the NA active site (Fig. 4d, e). Notably, the DR motif is reversed in order in FNI9, appearing as R–D instead of D–R, as seen in DA03E17 CDR H3 (Fig. 4i), yet both motifs mimic sialic acid interactions from nearly identical positions in the active site. In addition, we identified the same sialic acid-mimicking DR motif in the previously reported cross-group mAb Z2B3[47] (Fig. 4f), which binds to both N1 and N9 NAs. These findings highlight the conserved role of DR motifs in mediating receptor mimicry across multiple NA active site-targeting antibodies, supporting the idea of sequence and structural convergence in antibody responses to NA. Similarly, sialic acid mimicry has been previously reported in the N9 NA-specific mAb NA-45[46], where the negatively charged side chain of E (Glu), similar to D (Asp), is used to mimic the carboxylate group of sialic acid (Fig. 4g). This type of sialic acid mimicry, using D (Asp) and E (Glu) to mimic the carboxylate group of sialic acid, has also been observed in several HA receptor-binding site (RBS)-targeting antibodies[48–53], highlighting the critical role of negatively charged amino acids in mimicking the carboxylate group of sialic acid within both the HA RBS and the NA active site. We also compared

the structure of another pan-influenza NA mAb, 1G01[36], which has an R (Arg) at the tip of its CDR H3, but its interactions were distinct from those mediated by the DR motif (Fig. 4h). In summary, these NA active site-targeting antibodies possess relatively long CDR H3 loops (Fig. 4i), and while the Fabs have varying angles of approach and the CDR H3 loops engage the active site pocket in diverse ways (Fig. 4j, k), the DR motifs remain structurally conserved among antibodies with receptor mimicry.

## DR motif precursors are prevalent in human antibody repertoire

To assess the feasibility of eliciting NA active site-targeting antibodies that mimic sialic acid binding through vaccination, we employed a bioinformatic approach to identify the prevalence of antibodies with long CDR H3s containing DR motifs. Notably, in all three antibodies that had the DR motif and for which a D gene reading frame and position could be inferred (DA03E17, 1G05, and Z2B3), the DR motif was encoded at least partially by non-templated junction residues between the D and J genes (Fig. 5a, b), suggesting the DR motif could be limiting the precursor frequency. We employed two distinct search strategies using an ultradeep next-generation sequencing (NGS) dataset of $1.1 \times 10^9$ antibody heavy chain sequences from 14 healthy human donors[54,55]. First, we identified long CDR H3 loops (19–30 amino acids) with DR motif (DR or RD) positioned near the middle of the loop (flanked by at least 8 residues on either side) (Fig. 5c and Supplementary Fig. 10). CDR H3 loops bearing the DR motif were identified in all 14 donors, with median frequencies of 875 (DR) and 1348 (RD) per million sequences (Fig. 5d). No single D gene was dominant among these CDR H3 sequences. Given the potential for additional requirements, only a subset of precursors meeting this bioinformatic definition would be expected to have the potential to develop into NA active site-targeting antibodies. Therefore, this search represents an upper limit on the frequency of potential DR motif precursors that may be accessible to targeting by vaccination. The second search incorporated criteria that was specific for each of the three NA active site-targeting antibodies for which the D gene could be inferred (DA03E17, 1G05, and Z2B3). We defined precursors as having CDR H3 loops of equal length or longer than the template antibody (up to 30 amino acids) with the D gene in the same reading frame and flanked by at least as many amino acids as occurs in the template antibody (definition shown in Fig. 5e and in the "methods" section). Precursors for each antibody class were identified in all 14 donors at median frequencies of 1948, 228 and 6.9 per million for DA03E17, 1G05 and Z2B3, respectively (Fig. 5f). After establishing that all donors made precursors with these CDR H3 features, we then added a requirement for the DR motif to be present at the equivalent position as the template antibody relative to the D gene (Fig. 5e). Inclusion of the DR motif resulted in a large decrease in precursor frequencies for DA03E17 and Z2B3. DA03E17 precursors remained detectable in all 14 donors but at a 985-fold lower median frequency (~2 per million), whereas Z2B3 precursors were identified in only 2 of 14 donors, at a median frequency of ≤ 0.01 per million. In contrast, 1G05 precursors containing the DR motif occurred at a median frequency of 14 per million, representing only a 16-fold reduction likely due to D gene templating of the aspartic acid residue (Fig. 5f). These results suggest that targeting precursors with DA03E17- or 1G05-like CDR H3 loops with DR motifs in the naïve sequence may be feasible, as their precursor frequencies are comparable to other vaccine targets that have shown promising results[56–58]. Elicitation of Z2B3-like responses may require initial targeting of sequences lacking the DR motif and acquiring the DR motif through somatic hypermutation (SHM).

## Additional DR motif antibodies converge on the NA active site through receptor mimicry

To explore whether similar DR motif NA antibodies have been independently observed and to get a broader qualitative sense of their prevalence, we surveyed antibody sequences from patents and the literature for NA antibodies with a central DR motif in the CDR H3, thereby complementing our repertoire analysis. Using the Patent and Literature Antibody Database (PLAbDab)[59], we searched for antibody sequences using "neuraminidase" as a keyword and identified 168 paired antibody sequences derived from 24 different sources. Through manual curation, we identified two antibodies (CR12042 and CR12044) with a centrally positioned DR motif in the CDR H3, both of which were reported in a single patent[60] (Fig. 6a). In addition, we identified three more antibodies (AF9C, Z1A11, and Z2C2) from the study that also reported Z2B3[61], one of the DR motif antibodies included in our structural comparison (Fig. 4f). While CR12042 and CR12044 are described as human-derived antibodies, no details were provided regarding their source or method of isolation. CR12042 and CR12044 bound to multiple N1 NAs and showed weak binding to an N2 NA. Since they exhibited NI activity in a small molecule-based assay[60], we considered it likely that they target the NA active site. AF9C was isolated from an adult who received the 2014/15 Northern Hemisphere trivalent influenza vaccine (TIV), whereas Z1A11 and Z2C2 were isolated from a child who experienced a mild H7N9 infection in 2013, the same individual from whom Z2B3 had previously been isolated[61]. AF9C was N1-specific, while Z1A11 and Z2C2 were cross-reactive to N1 and N9 NAs, similar to Z2B3. Their NI activity was observed in ELLA, but was not assessed in a small molecule-based assay, leaving it unclear whether they directly target the NA active site. All five antibodies possessed a centrally located DR motif in the CDR H3 (Fig. 6a), and except for CR12042, which has an 18-residue CDR H3, they met the search criteria used for our precursor frequency analysis (Fig. 5c), which required a CDR H3 of 19–30 residues flanked by at least eight residues on either side of the DR motif. These five antibodies were expressed as IgGs, with AF9C, Z1A11, and Z2C2 produced in germline-reverted form (GL), as only gene usage and the CDR H3 and CDR L3 sequences were available. To evaluate their binding activity, we measured binding to CA09 N1 sNAp by BLI. Four of the five antibodies showed binding, with Z2C2-GL as the exception, possibly due to reduced affinity in its germline-reverted form lacking somatic mutations that may be required for stable interaction (Fig. 6b). To investigate whether the DR motifs in these antibodies mediate receptor mimicry in a structurally conserved manner, we determined cryo-EM structures of their complexes with CA09 N1 sNAp at resolutions ranging from 2.48 to 2.76 Å (Fig. 6c and Supplementary Fig. 11). All four antibodies targeted and blocked the NA active site, despite exhibiting distinct angles of approach (Fig. 6c). Remarkably, all DR motifs in these four antibodies were structurally conserved and engaged active site residues through receptor mimicry in a manner similar to the DR motif antibodies described earlier in this study (DA03E17, FNI9, 1G05, Z2B3) (Figs. 4 and 6d). These results further support the idea that receptor mimicry mediated by conserved DR motifs is a recurring strategy among NA active site-targeting antibodies.

We next investigated whether these eight DR motif antibodies share additional structural similarities beyond the DR motif. Although these antibodies exhibit diverse binding orientations to the NA active site (Fig. 6e), Z2B3 and AF9C-GL engage the active site with similar angles of approach (Supplementary Fig. 12). Both antibodies are derived from the *IGHV1-69*01* germline gene, but their binding angles are slightly offset, and no notable structural similarities were observed at the level of detailed interactions, with their light chains also originating from different germline genes (Z2B3 from *IGLV2-14*01* and AF9C from *IGKV1-9*01*)[61]. FNI9, CR12042, and CR12044 were also derived from, or predicted to originate from, the *IGHV1-69* germline gene (Figs. 5a and 6a). Although FNI9 and CR12042 approached the NA active site with similar Fv angles, the orientations of their heavy and light chains were reversed (Supplementary Fig. 12). CR12044 also bound with a distinct orientation. Thus, aside from the relatively frequent use of *IGHV1-69*, no major structural similarities were observed

**a**

| | | DA03E17 | FNI9 | 1G05 | Z2B3 |
|---|---|---|---|---|---|
| Antibody | | DA03E17 | FNI9 | 1G05 | Z2B3 |
| Antibody origin | | H1N1-infected patient in 2015-2016 season | Healthy individual | IBV-infected patient in 2017-2018 season | H7N9-infected patient in 2013 |
| Binding breadth | | IAV and IBV NAs | IAV and IBV NAs | IBV NAs | N1 and N9 NAs |
| HC germline | IGHV | 4-31*03 | 1-69*01 | 4-61*08 | 1-69*01 or 1-69D*01 |
| | IGHD | 3-10*01 | n/a | 5-18*02 | 5-18*01 |
| | IGHJ | 6*02 | n/a | 6*03 | 6*02 |
| CDR H3 sequence and length* | | ARVDGSGNT**DR**YY FYGMDV (19 aa) | ARAGSDYFN**RD**LG WENYYFAS (21 aa) | ARGDYSGY**DR**DV QVELMDV (19 aa) | ARDLQDTPMV**DR**IIGS YYYYNGLDV (25 aa) |
| Encoding region for the DR motif | | D-J junction | n/a | D gene and D-J junction | D-J junction |

*IMGT CDR definition used; n/a: Genetic sequence not available

**b**

**c**

**Search criteria:**

i. CDR H3 length: 19–30 amino acids

ii. DR or RD motif occurred approximately in the middle of the CDR H3 (≥ 8 amino acids on either side)

**d**

**e**

**Search criteria:**

i. Inferred D gene ± Sequence motif

```
DA03E17     (....GSG............)
DA03E17+DR  (....GSG..DR........)

1G05        (....YSGYD..........)
1G05+DR     (....YSGYDR.........)

Z2B3        (.....DT.MV.............)
Z2B3+DR     (.....DT.MVDR...........)
```

ii. Flanked by any number of amino acids up to a maximum CDR H3 length of 30 amino acids

**f**

**Fig. 5 | DR motif precursors are prevalent in humans. a** Genetic characteristics of DR motif antibodies targeting the NA active site through receptor mimicry. **b** The germline and amino acid sequences of DA03E17 CDR H3 are aligned, with non-templated junction residues in red and residues that have undergone SHM in light blue. Nucleotides removed by exonuclease trimming indicated with a line through the letters. Circled bullet points denote antibody main chain contacts only; open circles with rays denote antibody side chain contacts only; Circled bullet points with rays denote both main and side chain contacts. **c** Search criteria used to identify the minimal NA active site-targeting CDR H3 motifs in a NGS dataset of 1.1 billion human antibody heavy chain sequences. **d** Frequency of precursor BCRs with long CDR H3s containing DR (left) or RD (right) motifs near the middle of the CDR H3, displayed by total and the five D genes with the highest frequencies. Points represent the frequency observed in each of the 14 human donors. **e** Search criteria used to identify DA03E17-, 1G05- and Z2B3-like CDR H3s in the NGS dataset. **f** Frequency of precursors defined in (**e**) for three NA active site-targeting antibodies (DA03E17, 1G05, Z2B3). Points represent the frequency observed in each of the 14 human donors. Bars represent the median frequency across donors. Source data are provided as a Source Data file.

among these antibodies. Nevertheless, one notable commonality was the relatively frequent usage of the *IGHJ6* gene. DA03E17, 1G05, Z2B3, CR12042, AF9C, Z1A11, and even Z2C2, for which structural determination was not achieved due to loss of binding, all utilized *IGHJ6* (Figs. 5a and 6a). The *IGHJ6* gene is the longest among human IGHJ gene segments[62], and its increased usage has been observed among antibodies with long CDR H3 loops[63]. This may contribute to the maturation of antibodies capable of reaching into the deep NA active site pocket (Fig. 6f), thereby potentially facilitating the evolution of DR motif-bearing antibodies that mediate receptor mimicry. Together,

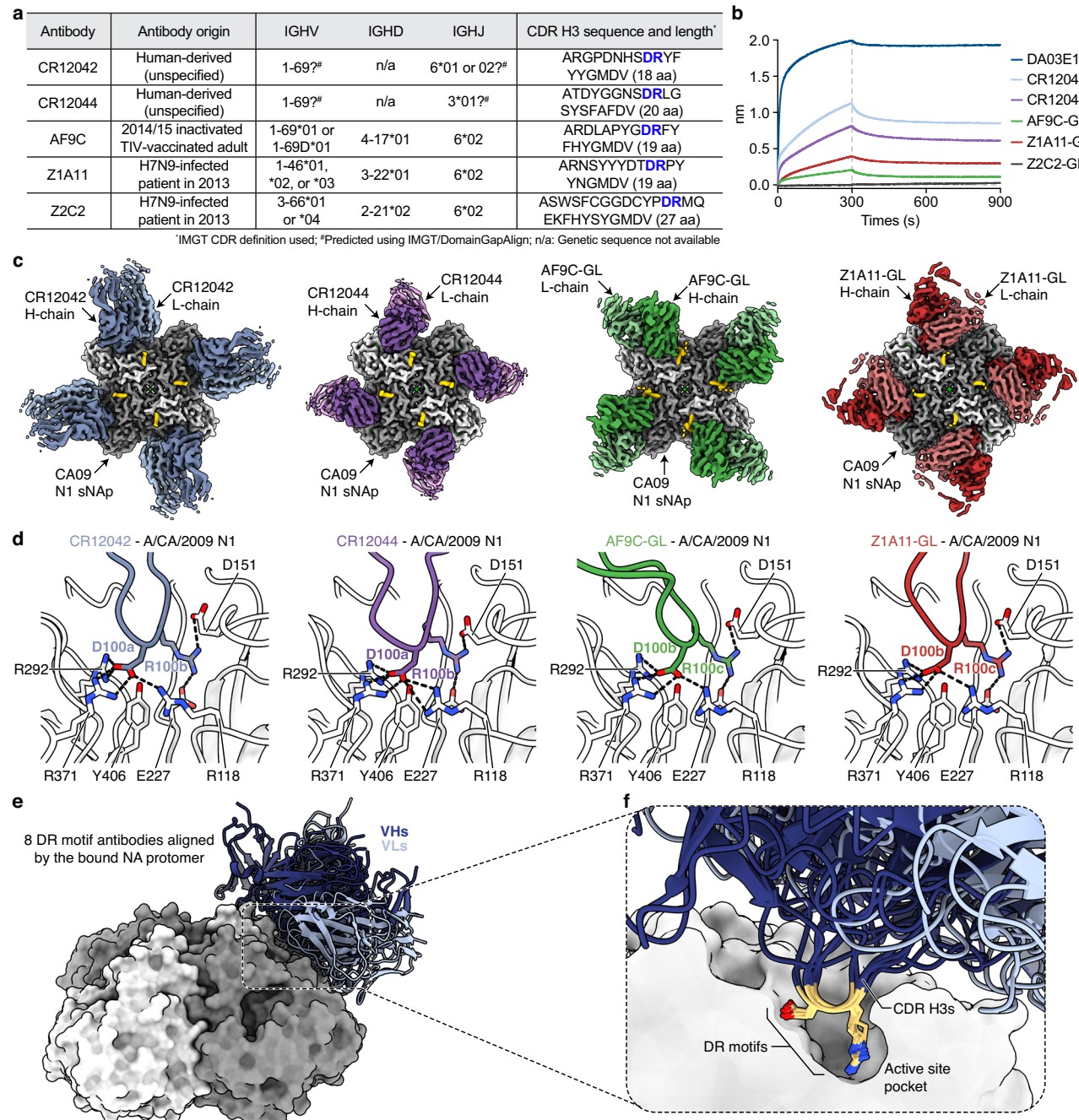

**Fig. 6 | Identification and structural characterization of additional DR motif NA antibodies. a** Genetic characteristics of additional DR motif antibodies. **b** Sensorgrams from BLI experiments showing binding of IgGs of CR12042, CR12044, and of germline-reverted (GL) AF9C, Z1A11, and Z2C2 to CA09 N1 sNAp immobilized on HIS1K biosensors. DA03E17 was included as a positive control. Source data are provided as a Source Data file. **c** Cryo-EM maps of additional DR motif antibodies in complex with CA09 N1 sNAp, with overall resolutions ranging

from 2.48 to 2.76 Å. **d** Detailed interactions between the conserved DR motifs in the CDR H3 of additional DR motif antibodies and the NA active site residues. **e** Structural alignment of 8 DR motif antibodies by their bound NA protomer. The variable regions of the heavy and light chains (VH and VL) are colored in dark blue and light blue, respectively. **f** Cross-sectional view of the NA active site showing conserved DR motifs occupying the active site pocket.

these findings expand the set of known DR motif antibodies, providing additional evidence for the structural and mechanistic convergence of this antibody class and highlighting its potential as a target for NA-based universal influenza vaccine.

## Discussion

While over 5000 human mAbs targeting HA have been identified[40], far fewer antibodies have been characterized for NA[28]. This imbalance underscores the need to continue isolating and studying NA-specific

antibodies to better understand the immune response to NA and identify broadly protective epitopes. Such efforts will reveal common molecular features of NA antibody responses that can inform the development of universal vaccines[28]. In this study, we provide a detailed structural and functional analysis of DA03E17, a broadly protective human mAb that targets the NA active site. Using cryo-EM, we determined the structures of DA03E17 bound to NAs from influenza A (H1N1, H3N2) and B viruses, revealing that DA03E17 inhibits the siali-dase activity of NA by blocking the highly conserved active site

through receptor mimicry using the conserved DR motif in CDR H3. The NA active site is highly conserved, and NA inhibition activity causes arrest of progeny virus egress from the host cell surface, preventing spread[64]. Conserved receptor mimicry strategies employed by several NA active site-targeting antibodies likely drive a key mechanism through which broadly protective antibodies inhibit viral spread. Therefore, eliciting antibodies via vaccination that target the NA active site through receptor mimicry holds strong potential for providing broad protection against diverse influenza strains.

DA03E17 retains its binding capacity even in the presence of oseltamivir-resistant mutations, particularly in N1 and N2 subtypes, further demonstrating its robust and broad reactivity. Furthermore, DA03E17 also binds to the NA of the bovine HPAI H5N1 virus currently spreading in U.S. dairy cattle and being transmitted to other mammals, including humans[11]. Considering its nanomolar binding affinity to this bovine H5N1 NA, along with its in vivo protective efficacy against the typical avian H5N1 virus (A/Vietnam/1203/2004) isolated from a human, as demonstrated in our previous study[38], DA03E17 could also serve as a valuable therapeutic option if this virus were to spread more widely in humans.

The ongoing evolution of H3N2 strains and the growing challenge of matching vaccines to circulating viruses[65] highlight the need for broadly protective antibodies that remain effective against these evolving strains. Our cryo-EM structural analysis and MD simulations revealed that DA03E17 accommodates the N245 glycan of drifted H3N2 viruses by inducing conformational changes in the glycan. Similarly, 1G01 also induces conformational changes in the glycan itself, while FNI9 induces additional structural rearrangements in the NA loop to accommodate the N245 glycan. These differences suggest that DA03E17 and 1G01 engage the N245 glycan with minimal perturbation to the NA loop, representing an alternative mode of accommodation distinct from that of FNI9. The ability of these antibodies to structurally accommodate the N245 glycan of drifted H3N2 viruses may be of particular relevance, as H3N2-dominated influenza seasons are often associated with greater disease burden and higher hospitalization rates, especially in vulnerable populations[66]. While these structural observations highlight varying modes of glycan engagement, further experimental data would be needed to determine whether these differences translate into functional advantages. Despite the broad cross-reactivity of DA03E17, we observed substantially reduced binding to recent N2s such as A/Kansas/14/2017, suggesting that drifted N2s may pose a challenge for NA active site-targeting antibodies. Future strategies could include immunogen designs that focus immune responses on conserved regions of N2 or structure-guided engineering of existing antibodies to improve their compatibility with recently evolved N2 NAs.

Our structural and immunogenetic analyses suggest that long CDR H3s with the DR motif near the middle of the CDR H3 could be an important requirement for effectively targeting the NA active site. Bioinformatic analysis using ultradeep NGS data shows that potential precursors with these features are prevalent in the human antibody repertoire. Furthermore, a comparison of three specific NA antibodies that use the DR motif shows their precursor frequencies span a wide range in the human antibody repertoire, from undetectable to ~1 in $10^5$. These findings highlight the importance of selecting the right targets for vaccine design if these antibodies are to be consistently elicited by vaccination. The antibodies compared here have varying angles of approach and peripheral contacts, but they all bind the active site through DR motif-mediated receptor mimicry, maintaining broad reactivities. In addition, they have diverse germline origins, with *IGHJ6* being the only shared feature, indicating their potential to be broadly elicited across human populations. The accompanying manuscript by Lederhofer and Borst et al. further reinforces these findings, presenting additional human antibodies with DR motif-mediated receptor mimicry, thereby underscoring the prevalence of DR motif antibodies

in humans and providing structural and functional insights through both in vitro and in vivo evaluations. Nevertheless, while the prevalence of potential precursors is promising, the ability of these precursors to mature into functional broadly protective antibodies remains to be fully understood.

In conclusion, our study emphasizes the importance of long CDR H3s with the DR motif for targeting the NA active site. The structural insights gained from DA03E17, combined with the investigation of DR motif precursors in the human antibody repertoire, provide a strong foundation for developing vaccines that can elicit broadly protective antibodies. Utilizing this molecular signature to interrogate immune responses to NA and design immunogens capable of triggering protective responses across diverse populations, therefore has the potential for mitigating future influenza outbreaks.

## Methods

### Expression and purification of recombinant NAs and antibodies

The NAs of the H1N1 A/California/07/2009 (CA09), H1N1 A/Brisbane/02/2018 (BB18), H5N1 A/dairy cattle/Texas/24-008749-001/2024 (TX24), H3N2 A/Perth/16/2009 (PT09), H3N2 A/Indiana/08/2011 (IN11 N2), H3N2 A/Kansas/14/2017 (KS17), H3N2 A/Hong Kong/2671/2019 (HK19), and B/Colorado/06/2017 (B/Victoria-lineage; CO17) were expressed in the mammalian expression system as previously described[41]. Briefly, the ectodomain heads of the IAV N1 (residues 83–470 in N2 numbering), N2 (residues 83–469 in N2 numbering) NAs, and CO17 B (residues 76–466) NA were fused to an N-terminal IL2 signal peptide, a hexahistidine tag, a Strep-tag, a vasodilator-stimulated phosphoprotein (VASP) tetramerization domain, and a thrombin cleavage site. Of note, all N1 NA constructs have previously reported stabilizing mutations in the inter-protomeric interface for the closed tetrameric state (stabilized NA protein, sNAp)[41]. The CA09 N1 sNAp contains ten stabilizing mutations: I99P, Y100L, C161V, E165S, S171A, V176I, S195T, V204I, R419V, and Q412M (N2 numbering). BB18 N1 sNAp and TX24 N1 sNAp also include these ten mutations but carry an additional T131Q substitution. NA constructs were transiently expressed in Expi293F cells (Thermo Fisher Scientific) at 37 °C, shaking at 125 rpm. The secreted recombinant NA proteins were purified from culture supernatant by Ni-NTA affinity chromatography (Qiagen) followed by size exclusion chromatography using a Superdex 200 Increase 10/300 column (GE Healthcare) in 1X Tris-Buffered Saline (TBS) pH 7.4. The purified NAs were quantified by optical absorbance at 280 nm, and purity and integrity were analyzed by reducing and nonreducing SDS-PAGE. For DA03E17 Fab expression in mammalian cells, the heavy and light chain variable regions of DA03E17 were cloned into the pAbVec containing the corresponding human $C_H1$ region of human IgG1 and kappa $C_L$ region, respectively. The Fab was transiently expressed in Expi293F cells (Thermo Fisher Scientific) at 37 °C, shaking at 125 rpm. The secreted recombinant Fab was purified from culture supernatant by CaptureSelect IgG-CH1 Affinity Matrix (Thermo Fisher Scientific), followed by size exclusion chromatography using a Superdex 200 Increase 10/300 column (GE Healthcare) in 1X Tris-Buffered Saline (TBS) pH 7.4.

### ELISA

To evaluate the binding of mAbs, 96-well ELISA plates (PerkinElmer) were coated overnight at 4 °C with 2 μg/ml of recombinant NA proteins. Plates were washed three times with a PBS solution containing 0.05% Tween-20 (PBS-T) and then blocked with 5% Non-fat milk (RPI M17200-500.0) in PBS-T. The plates were incubated for 1 h at room temperature and then washed three times with PBS-T. In the meantime, mAbs were diluted in PBS-T, starting at a concentration of 100 μg/ml with a threefold serial dilution, and then added to the plate for 1 hour at room temperature. After washing, HRP-conjugated goat anti-human IgG Fc secondary antibody (Southern Biotech) was added at a dilution of 1:20,000 in PBS-T and incubated for an additional hour

at room temperature. Plates were washed three times with PBS-T and then developed with TMB substrate (Thermo Fisher Scientific). The reaction was stopped by the addition of 2 N $H_2SO_4$, and absorbance was measured at 450 nm. The data is shown as one representative biological replicate with the mean ± SD for one ELISA experiment. The ELISAs were repeated 2 times.

## ELLA

96-well high-binding plates (SpectraPlate-96HB) were coated with 100 μL per well of 25 μg/mL fetuin (Sigma #3385) diluted in coating buffer (KPL #50-84-01). Coated plates were stored at 4 °C for up to two months before use. To determine the optimal concentration of recombinant NA for the assay, 50 μL of serially diluted recombinant NA (starting from 20 μg/mL with 1:3 dilutions) in sample diluent (PBS supplemented with $CaCl_2$ and $MgCl_2$, 1% BSA, and 0.5% Tween-20) was added to the fetuin-coated plates after three washes with PBS-T. Plates were sealed and incubated either at 37 °C for 16–18 h or at 33 °C for 6 h. After incubation, plates were washed five times with PBS-T and incubated with 100 μL of 1 μg/mL peanut agglutinin (PNA)-HRP (Sigma #7759-1 mg) diluted in sample diluent without Tween-20 for 2 h at room temperature. Plates were washed three times with PBS-T, and color development was performed using TMB substrate (Seracare #5120-0047). Data were analyzed using GraphPad Prism (version 10), and the linear portion of the NA titration curve was used to determine the concentration yielding a near-maximal $OD_{450}$ signal (at least 10-fold over background), which was used in subsequent inhibition assays. To assess NA inhibition, antibodies were serially diluted (1:2, starting from 20 or 30 μg/mL) in sample diluent and mixed 1:1 with recombinant NA at the predetermined working concentration. A total of 50 μL of the antibody-NA mixture was added to each well of washed fetuin-coated plates. The remaining assay steps were performed as described above. Percent inhibition was calculated as: 100 - (sample $OD_{450}$ / NA-only $OD_{450}$ × 100), where the NA-only control (no antibody) represented 0% inhibition. Midpoint inhibitory titers ($IC_{50}$) were determined using GraphPad Prism. All measurements were performed in duplicate.

## Biolayer interferometry (BLI)

Antibody affinities were determined by biolayer interferometry using an Octet Red96 (Sartorius). HIS1K biosensors (Sartorius) were hydrated in kinetics buffer (1 x PBS, pH 7.4, and 0.002% Tween 20) prior to use. Recombinant NA proteins (10 μg/ml) in kinetics buffer were immobilized on hydrated HIS1K biosensors for 300 sec through their hexahistidine tag at their N-termini, and baseline was measured in kinetics buffer for 120 sec. Following baseline measurements, the sensors were loaded with DA03E17 IgG or Fab (3-fold serially diluted from 450 nM or 2400 nM, respectively), and association and dissociation was measured for 300 and 600 sec, respectively. All assay steps were performed at 30 °C with agitation set at 1000 rpm. Baseline correction was carried out by subtracting the measurements recorded for a sensor loaded with the corresponding NA in the same buffer with no Fab. The data were analyzed using the Octet-Red96 software, and the association and dissociation rates were calculated using a 1:1 model with global curve fitting.

## Cryo-EM sample preparation and data collection

For each DA03E17 Fab complex with CA09 N1, KS17 N2, and CO17 B NAs, DA03E17 Fab was added at a 1:3 molar ratio of NA protomer to Fab and incubated at room temperature for 1 h before grid preparation. For the 1G01 IgG-KS17 N2 NA and FNI9 IgG-KS17 N2 NA complexes, each IgG was added at a 1:1 molar ratio of NA tetramer to IgG and incubated at room temperature for 1 h before grid preparation. For the additional DR motif NA antibodies shown in Fig. 6 (CR10242, CR12044, AF9C-GL, and Z1A11-GL), each IgG was added at a 1:2 molar ratio of NA tetramer to IgG and incubated at room temperature for 1 h before grid

preparation. Final protein concentrations were between 0.4-0.5 mg/ml. For DA03E17-NA complexes, samples were mixed with octyl-beta-glucoside (OBG; final concentration 0.1% w/v) detergent to aid in particle tumbling and applied to Quantifoil 1.2/1.3 300 mesh copper grids. The grids were plunge-frozen using a Vitrobot Mark IV (Thermo Fisher Scientific) with a blot force of 1 and a blot time of 3.5–5 sec at 100% humidity and 4 °C. For 1G01-KS17 N2 NA and FNI9-KS17 N2 NA complexes, samples were also mixed with OBG detergent but applied to Quantifoil 2/1 400 mesh copper and UltrAuFoil 1.2/1.3 300 mesh gold grids. The grids were plunge-frozen at room temperature with 100% humidity, a blot force of 1, and a blot time of 3.5–4.5 sec. For CR10242, CR12044, AF9C-GL, and Z1A11-GL complexes, samples were mixed with OBG detergent and applied to UltrAuFoil 1.2/1.3 300 mesh gold grids. The grids were plunge-frozen at room temperature with 100% humidity, a blot force of 1, and a blot time of 2–3 sec. All datasets were collected on a 200 kV Glacios (Thermo Fisher Scientific) equipped with a Falcon IV direct electron detector. Automated data collection was carried out using EPU (Thermo Fisher Scientific) at a nominal magnification of 190,000 × and a pixel size of 0.725 Å, with an approximate exposure dose of 45 e⁻/Å² and a nominal defocus range of − 0.7 to − 1.4 μm. For each dataset, between 1684 and 6416 movie micrographs were collected. For the DA03E17 Fab + KS17 N2 NA dataset, two rounds of data collection resulted in 3166 and 3700 movie micrographs, which were combined for processing.

## Cryo-EM data processing

All datasets were processed using cryoSPARC[67]. Dose-weighted movie frame alignment was carried out using Patch motion correction in cryoSPARC live to account for stage drift and beam-induced motion. The contrast transfer function (CTF) was estimated using Patch CTF in cryoSPARC live. Micrographs were curated based on CTF fits, with those worse than 6–8 Å excluded due to poor quality. Individual particles were selected from an initial subset of several hundred micrographs using Blob picker. After several rounds of 2D classification, classes resembling the Fab-NA complex were used to template pick for all micrographs. Clean particle stacks were selected through multiple rounds of 2D classification and used to generate a reference volume via ab initio reconstruction, which was then utilized as an initial volume for homogeneous refinement and/or non-uniform refinement. For the DA03E17-CA09 N1 sNAp and DA03E17-KS17 N2 NA datasets, Fab-NA complex particles underwent 3D classification. The best 3D classes were refined using either local refinement with a Fab-NA mask (DA03E17-CA09 N1) or non-uniform refinement (DA03E17-KS17 N2). Final maps were generated after global CTF refinement and application of C4 symmetry. For the apo-KS17 N2 NA, only NA particles without visible Fab binding were selected through multiple rounds of 2D classification, followed by non-uniform refinement with C4 symmetry applied and global CTF refinement, yielding the final map. For the DA03E17-CO17 B NA dataset, Fab-NA complex particles were subjected to homogeneous refinement with C4 symmetry applied, followed by global CTF refinement to generate the final map. For the 1G01-KS17 N2 NA dataset, particles selected after 2D classification were subjected to heterogeneous refinement. The best 3D classes were refined using local refinement with a Fab-NA mask and C4 symmetry applied, yielding the final map. For the FNI9-KS17 N2 NA dataset, particles selected after 2D classification were subjected to ab initio reconstruction with five classes. Particles from the highest-quality classes were then used for homogeneous refinement and local refinement with a Fab-NA mask, with no symmetry applied, yielding the final map.

## Cryo-EM model building and refinement

For model building, the structures of CA09 N1 NA (PDB: 6Q23), HK19 N2 NA (PDB: 8G3O), and PK13 B NA (PDB: 6V4N) were used as the initial models for CA09 N1 sNAp, KS17 N2 NA, and CO17 B NA, respectively. The DA03E17 Fv model was generated using ABodyBuilder2[68]. The

models were fitted into the cryo-EM maps using UCSF ChimeraX[69]. The models were manually adjusted using Coot[70] and further refined through Rosetta Relax[71] and real-space refinement in Phenix[72]. N1 and N2 NAs were numbered according to the N2 numbering scheme, while the antibody Fv was numbered based on the Kabat numbering scheme. Epitope and paratope residues, as well as their interactions, were identified by using the PISA server[73] with buried surface area (BSA > 5 Å$^2$) as the criterion. Structural figures were prepared using UCSF ChimeraX.

## NA Sequence conservation analysis

For Supplementary Table 2, NA sequences for IAV Group 1 (N1, N4, N5, and N8), Group 2 (N2, N3, N6, N7, and N9), and IBV (Victoria and Yamagata) were retrieved from GISAID. Sequences for each subtype were aligned using GENETYX ver13, and SNP analysis was conducted using the BV-BRC platform (https://www.bv-brc.org/). For Fig. 2, full-length NA protein sequences from human IAV H1N1, H3N2, and IBV B/Victoria- and B/Yamagata-lineage viruses circulating from 1977 to 2023 (H1N1), 1968 to 2023 (H3N2), 1987 to 2023 (B/Victoria), and 1988 to 2020 (B/Yamagata) were downloaded from GISAID. To avoid temporal sampling bias, we sampled at most 10 sequences per year, resulting in a total of 340, 421, 278, and 251 sequences for N1, N2, B/Victoria, and B/Yamagata NAs, respectively. Multiple sequence alignments were performed using Clustal Omega, and the reference sequences used for alignment were H1N1 A/California/04/2009, H3N2 A/Kansas/14/2017, and B/Colorado/06/2017. Sequence logos were generated using WebLogo 3 and manually curated in Adobe Illustrator. For Supplementary Fig. 7, we additionally analyzed animal-origin IAV N1 and N2 NA sequences derived from HxN1 and HxN2 viruses that have circulated in avian and mammalian hosts between 1977 and 2023. A total of 432 N1 and 404 N2 sequences were included. Representative subtypes included H1N1, H5N1, and H6N1 for N1, and H1N2, H2N2, H3N2, H4N2, H5N2, H6N2, H7N2, H9N2, H11N2, and H13N2 for N2. Epitope residues in all analyses were defined based on structural contacts observed in the DA03E17-CA09 N1 NA, DA03E17-KS17 N2 NA, and DA03E17-CO17 B NA complexes.

## Molecular dynamics

As starting models for our MD simulations, the cryo-EM structures of DA03E17-KS17 N2 NA (this study) and apo-KS17 N2 NA (this study) were used. The starting structures for our simulations were prepared in Molecular Operating Environment (MOE, CCG)[74] using the Protonate3D tool[75]. To neutralize the charges, the uniform background charge was applied, which is required to compute long-range electrostatic interactions[76]. Using the tleap tool of the AmberTools22[77] package, the structures were soaked in cubic water boxes of TIP3P water molecules with a minimum wall distance of 12 Å to the protein[78–80]. For all simulations, parameters of the AMBER force field 19SB were used[81]. For the glycans, we used the most recent GLYCAM force field, namely the GLYCAM-06j parameter set[82]. We then performed 3 repetitions of 1 µs of classical molecular dynamics simulations for each system. Molecular dynamics simulations were performed in an NpT ensemble using pmemd.cuda[83]. Bonds involving hydrogen atoms were restrained by applying the SHAKE algorithm[84] allowing a time step of 2 fs. The Langevin thermostat was used to maintain the temperature during simulations at 300 K[85,86] with a collision frequency of 2 ps$^{-1}$ and a Monte Carlo barostat[87] with one volume change attempt per 100 steps. Clustering analysis has been performed using the hierarchical average linkage clustering implement in cpptraj, by aligning on the Cα-positions of the NA and clustering on the glycans using a RMSD distance cut-off criterion of 2.5 Å[88].

## Antibody precursor frequency estimates

Precursor searching and frequency estimates of NA active site-targeting antibodies with receptor mimicry were performed using

methods described previously[57]. The precursor definition for the first search strategy had two requirements: (1) CDR H3 length ranging from 19 to 30 amino acids, and (2) the DR motif occurred approximately in the middle of the CDR H3, ≥ 8 amino acids on either side. These criteria were based on the CDR H3 length and DR motif position in several NA active site-targeting antibodies. These definitions allowed for precursors with long CDR H3s possessing a DR motif, regardless of germline gene usage, with diverse V-D and D-J junctions. A total of $1.1 \times 10^9$ BCR heavy chain sequences were searched in a ultra-deep NGS dataset of 14 human donors[54,55]. For the second strategy, we searched for precursors to three antibodies (DA03E17, 1G05, and Z2B3) using the following criteria: (1) CDR H3 length ≥ the target NA antibody. (2) D gene in the same reading frame as the NA antibody and matching the regular expression shown in Fig. 5e. (3) The D gene is flanked by the same number or more amino acids compared to the target NA antibody. (4) The DR motif is at the same position relative to the D gene as in the target NA antibody. The following D genes were used as search criteria: DA03E17 (D3-10), 1G05 (D5-12), and Z2B3 (D5-18/D5-5). While both 1G05 and Z2B3 are predicted to use D5-18, they use two different alleles (D5-18*01 and D5-18*02), which have substantially different nucleotide sequences. Also, D5-18*02 is only different from D5-12 by one nucleotide. Because D5-18*02 was not in our NGS dataset, we searched for 1G05 precursors using D5-12 instead. For Z2B3 precursors, we searched using both D5-18*01 and D5-5, as these D genes are identical.

## Reporting summary

Further information on research design is available in the Nature Portfolio Reporting Summary linked to this article.

## Data availability

The atomic models and cryo-EM density maps generated in this study have been deposited to the Protein Data Bank (PDB) and Electron Microscopy Databank (EMDB), respectively. The accession codes are 9CYE and EMD-46041 (DA03E17 Fab + CA09 N1 NA), 9CYF and EMD-46042 (DA03E17 Fab + KS17 N2 NA), 9CYG and EMD-46043 (Apo-KS17 N2 NA), 9CYH and EMD-46044 (DA03E17 Fab + CO17 B NA), 9CYI and EMD-46045 (1G01 IgG + KS17 N2 NA), 9CYJ and EMD-46046 (FNI9 IgG + KS17 N2 NA), 9O4N and EMD-70108 (CR12042 IgG + CA09 N1 NA), 9O4O and EMD-70109 (CR12044 IgG + CA09 N1 NA), 9O4P and EMD-70110 (AF9C-GL IgG + CA09 N1 NA), and 9O4Q and EMD-70111 (Z1A11-GL IgG + CA09 N1 NA). Source data are provided in this paper.

## Code availability

PySpark scripts used in precursor frequency analysis are available at https://github.com/SchiefLab/Jo2025, along with instructions on setting up an EMR cluster.

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

## Acknowledgements

We thank Hannah Turner, Anant Gharpure, and Will Lessin for electron microscopy support. We thank Charles Bowman and J.C. Ducom for computational support. We thank Lauren Holden for administrative support. This research was supported by the NIH National Institute of Allergy and Infectious Diseases (NIAID) Collaborative Influenza Vaccine Innovation Centers (CIVICs) contract grant 75N93019C00051 (to A.B.W. and Y.K.), NIH grant R21-AI185735 (to J.M.S.), Third Rock Ventures (to J.L.T., J.A.F., M.L.F.Q., J. Huang, A.J.R., J. Han, and A.B.W.), the Japan Agency for Medical Research and Development (JP24wm0125002 and JP243fa627001 to Y.K.), and the Basic Science Research Program through the National Research Foundation of Korea (NRF) funded by the Ministry of Education (RS-2023-00240483 to G.J.).

## Author contributions

J. Han and A.B.W. conceptualized the studies. G.J., J. Huang, J.C., and A.J.R. expressed and purified proteins. G.J., J.L.T., J.A.F., and J. Han prepared cryo-EM samples and collected and processed cryo-EM data. G.J. built and refined atomic models. G.J. and J. Huang performed

biochemical experiments. G.J., S.Y., and O.S. conducted the NA sequence conservation analysis. J.M.S. and K.M.M. performed the precursor frequency analysis. M.L.F.Q. performed molecular dynamics simulations. G.J., J. Han, J.M.S., K.M.M., S.Y., and M.L.F.Q. wrote the original manuscript draft with input and edits from Y.K., J. Han, and A.B.W. All authors contributed to the manuscript review and editing.

## Competing interests

J.L.T., J.A.F., M.L.F.Q., J. Huang, A.J.R., J. Han, and A.B.W. conduct sponsored research for Third Rock Ventures. The remaining authors declare no competing interests.
