## [Transparent Peer Review file · Nature Communications]

Structural basis of broad protection against influenza virus by human antibodies targeting the neuraminidase active site via a recurring motif in CDR H3

Corresponding Author: Dr Andrew Ward

Version 0:

Reviewer comments:

Reviewer #1

(Remarks to the Author)

In this study Gyunghee and co-authors present the cryo-electron microscopy structures of the broadly IAV and IBV NA-inhibiting antibody DA03E17, which was previously isolated by the same group from an individual infected with A/H1N1pdm09 during the 2015–2016 influenza season. Similarly to other broadly NA-targeting monoclonal antibodies, DA03E17 binds to the highly conserved NA active site using its long CDR3 loop and mimicking the interaction of the sialic acid receptor. The authors reveal that an Asp–Arg (DR) motif, present in the CDRH3 of different broadly NA-inhibiting mAbs, represents the basis of this receptor mimicry and the broad activity against seasonal IAVs and IBVs. Moreover, this motif enables binding to oseltamivir-resistant and zoonotic IAV strains, including the currently circulating H5N1, thus highlighting its importance and in general the importance of this class of antibodies for universal influenza vaccine design. Overall, the study presents novel structural information related to DA03E17 and other previously reported broadly NA-targeting mAbs and provides valuable insight on this broad activity and the possibility to elicit this class of antibodies through vaccination.

Major comments

Extended Figure 1b

1G05 was included as negative control since it binds to IBV NAs, but it would be important to have an idea of how DA03E17 activity against oseltamivir-resistant mutants compares with other previously described broadly NA inhibiting mAbs (e.g. 1G01 or FN19). Considering that the authors presented structural data for 1G01 and FN19, test these mAbs side by side should not be a problem.

Although some of the oseltamivir-resistant mutations are associated with reduced neuraminidase activity, for those sNAP displaying enough sialidase activity the authors should produce ELLA results beside ELISA binding data.

Extended Data Figure 3

Do the authors have any thought on the reason why DA03E17 have a significant lower affinity against N1 from H5N1 dairy cattle in comparison to the N1 from H1N1 CA09.

Could the authors add the affinity of DA03E17 to the sNAP bearing oseltamivir-resistant mutations used for the ELISA assay reported in Extended Fig. 1. This would be helpful to have an idea of the affinity against the parental N1 or N2 and loss against ose-resistant mutations.

Figure 2

Any analysis to assess the conservation also across IAV NAs from animal circulating IAVs (mammals or avian species)?

Despite the B-Yamagata-like lineage is considered extincted, it would be important to mention at least in the text if the

authors observed any difference in conservation of the DA03E17 epitope between the NAs of Victoria-like and Yamagata-like lineages at least up to 2020?

Figure 2 panel d:

Based on lines 148-150 in the text: Shouldn't all the 19 NA residues contributing the most to the DA03E17 binding be denoted with symbols above the logoplots?

Along the same line Y406, I222, N294 are described as relevant catalytic site residues for DA03E7 CDRH3 binding to N1, but they are not indicated by any circle or circle with rays in the logoplot. If these amino acids are not so relevant, maybe it would be better to indicate in the text (148-150) only those residues that have crucial impact on DA03E17 binding.

Related to Fig.5

The analysis of the DR motif prevalence in the human antibody repertoire appears quite limited and bit confused. Paragraph 296-309 is confusing as the authors move from the description of the deep sequencing dataset to state the structural characteristics of the previously described NA active site-targeting mAbs. Suggestion: move lines 296-298 describing the NGS approach and the number of human donors after the consideration on the broadly active mAbs and right before lines 307-309.

Considering that 2 out of 4 mAbs presented in panel Fig.5a use IGHD5-18, it would be interesting to have an idea of the frequency of the BCR precursor also for the IGHD5 gene or at least to comment on that.

Discussion

Lines 334-337: "Considering its strong binding affinity to this bovine H5N1 NA...". This sentence appears an overstatement considering the affinity reported for DA03E17 in Extended Data Figure 3 against N1 from dairy cattle in comparison to N1 from Cal09.

Lines 344-345: "DA03E17 and 1G01 may have the advantage in that they can accommodate this glycan without inducing structural changes in the NA loop, allowing for more stable and efficient binding". This statement is not supported by any data. Furthermore, based on Momont et al. 2023, in neutralization tests performed against recent H3N2 bearing N245 glycan FNI9 was consistently superior to 1G01 in its activity and 1G01 affinity was impacted to a higher extent than FNI9 by the 245 glycan. In addition, DA03E17 affinity is strongly reduced against N2 from H3N2 A/Kansas/2017 and Hong Kong /2019 in the Extended Data Figure 3. The authors should support this statement with *in vitro* tests comparing the activity of DA03E17 and 1G01 with FNI9 against N2 bearing N245 glycan or they should tune down this statement.

Minor comment

Fig.1h: labels for amino acids belonging to CDRL1, L3, and FR L3 are very difficult to read when printed on paper. Suggestion: use a darker border color.

Reviewer #2

(Remarks to the Author)

Jo et al. describe the molecular structure and binding epitope of the broadly protective human antibody DA03E17, which is directed against the enzymatic active site on NA, and elucidate the mechanism on how this mAb achieves broad reactivity against NA from A/H1N1, A/H3N2, and B/Victoria-lineage viruses. The same group had described this human mAb in a prior publication demonstrating the anti-viral effectiveness in mice against multiple Influenza strains (Yasuhara, A. et al. A broadly protective human monoclonal antibody targeting the sialidase activity of influenza A and B virus neuraminidases. *Nat Commun* 13, 6602-6655 (2022)).

In general, the manuscript is well written, and the data are of interest to the scientific community because it identifies the structural characteristics of this mAb, which enables its binding directly into the enzymatic pocket of NA, and how it accommodates the N-glycans of drifted N2 NAs. The authors show that this ability to accommodate the N-glycans of drifted N2 NAs seems to be a common feature with additional NA active site targeting mAbs the authors evaluated. The authors also identified a motif that is conserved among several mAb NA active site targeting antibodies, indicating a common receptor mimicry strategy.

Comments the authors may want to consider:

- Trying to determine the prevalence of DA03E17-like mAb in the human antibody repertoire, the authors utilized an ultradeep next-generation sequencing (NGS) dataset from 14 healthy human donors for the precursor frequency search. By using the DR- or RD-motif, which seems to be common between six NA active site-targeting antibodies, and the characteristic long CDR H3 loop found in those mAbs, the authors identified frequencies of such BCR sequences of 1 in 1,143 and 1 in 742, respectively, in all 14 donors. The authors acknowledge that the proportion of these DR or RD motif BCRs capable of maturing into functional NA antibodies remains to be determined, but it seems a little stretched to assume that just having a long CDR H3 loop with the DR or RD-motif is sufficient to allow the generation of these kinds of mAbs. Therefore, the statement that the relatively high precursor BCR frequency the authors found with this method is "highlighting the prevalence of this motif and its potential as vaccine targeting" seems a big stretch in my opinion. Not only is it unlikely that the precursor BCR frequency identified by the authors really is representative of NA-specific mAb, but identifying

strategies to expand those precursors and to induce large frequencies of NA active site-targeting antibodies through vaccination will be a significant challenge. This should be discussed more thoroughly on how the knowledge gained by the authors would allow the induction of DA03E17-like mAb. The authors may want to consider toning down this statement in the abstract.

- In addition to the DR- or RD-motive, would it be possible, based on the structural constraints of how those mAb can bind into the enzymatic pocket, to determine additional anchor amino acid requirements to refine the BCR-screening criteria and maybe determine a more realistic frequency of similar BCR, with potential of binding to NA, in the human repertoire?

- In order to provide more information on how frequent such NA active site-targeting mAb are after infection, performing competition ELISAs between the serum from those subjects used to identify the NA active site-targeting antibodies and the mAb DA03E17 would be informative. Performing direct NI inhibition in the MUNANA assay with the serum could also provide information about the contribution of NA active site-targeting antibodies in the serum to NI activity. I think it would be of high interest to the scientific community to understand how prevalent such Abs are in serum from subjects after infection, not just as potential pre-cursor. That would provide a baseline which next generation vaccines (NA+HA) would have to reach to induce similar protection. Not too many of these NA active site-targeting antibodies have been identified in humans (so far, unless the accompanying manuscript by Lederhofer and Borst et al. describes additional, similar mAb against NA?; I don't have access to it), which would imply that the induction of such Ab even after infection is not very frequent. Knowing how frequent the mAb DA03E17 (or the other, similar mAbs) was in the person from which it was isolated would further increase the scientific value of this manuscript.

- In the prior publication in which the mAb DA03E17 was first described (Yasuhara, A. et al. 2022), an escape mutation in NA at position T439A was identified. In this new manuscript, T439 was no longer identified as an important anchor. The authors may want to consider discussing how the T439A substitution fits with their structural model on how the mAb DA03E17 binds to NA.

- Line 213: the N146 glycan substitution is mentioned here for the first time in the manuscript. Could the authors mention when this N146 glycan substitution was first seen in circulating NA strains?

Minor corrections:

- Line 421: change "30 C" to "30°C"

Reviewer #3

(Remarks to the Author)

The manuscript by Jo et al. builds upon three recently published studies (DOI: 10.1126/science.aay0678, DOI: 10.1038/s41586-023-06136-y, and DOI: 10.1038/s41467-022-34521-0) that focused on isolating and characterizing broadly inhibiting anti-NA antibodies from humans. This study combined new structural insight from the binding of the human antibody DA03E17 with the other available NA inhibitory antibody binding structures to identify a DR (Asp-Arg) motif in the CDR H3 that has the potential to mimic NA substrate interactions like those observed in DA03E17, resulting in the reported broad binding capacity against NAs. The prevalence of this motif was then screened in a healthy human donor BCR database suggesting it is commonly present in human and vaccines can potentially be developed to target these BCRs to elicit this broad class of anti-NA antibodies. Together, these results support the findings from previous studies that isolated this class of antibody against NA from different sources. It also speculates that vaccines could be developed to elicit these antibodies, supporting the concept that vaccines containing NA could provide increased breadth against circulating strains. Overall, this is a well written manuscript with clear data followed by the identification of a speculative motif that would be of broad interest with the minor revisions outlined below that include the need for more experimental or computational support for the putative DR motif and updating the introduction.

Main points

1. This study builds upon, and substantially benefits from, prior published data showing NA inhibitory antibodies mimic the binding of the sialic acid substrate. In its current form the introduction does not match the study and should be rewritten to focus more on NA and include the highly related and supportive published literature that this study is dependent on for identifying the putative DR/RD motif. This should strengthen rather than detract from the novelty of the study as it will increase the readability and support the more generalized statements about the mechanisms of action and binding of this class of antibodies. References should also be more explicit throughout the results section where appropriate.

2. The DR/RD motif search for potential precursors in human antibody repertoires presented in figure 5 is currently very speculative and would benefit from more thorough analysis to put the data in context for the reader. For instance, some experimental data generating and testing the NA inhibitory nature of a few DR and RD motifs either by transfer to existing antibodies or as peptides could be performed to demonstrate if the motif is dependent or independent of the antibody context in which it resides given it appears to be a protruding domain that can potentially function independently. Alternatively, a more thorough bioinformatic analysis could be performed such as what is the prevalence of the DR motif in the subset of the BCR CDRH3 sequences with respect to a series of 2 random amino acid combinations or rank order the DR motif versus all other 2 amino acid combinations. What if any sequence conservation is present in the left and right sides of the sequences with the DR motif? Did any of the DR motifs in the database match any of the isolate antibodies? As it stands now, the reader has no way to know what the frequencies of 1 in 1,143 and 1 in 742 means and the conclusion is highly speculative.

3. The KD calculations are all performed with IgG, which has a large avidity effect due to the two Fab arms. The authors should either repeat these measurements with FAbs or be explicit about the IgG avidity in the text and use 'apparent' for all KD measurements.

Minor points

1. Extended data fig 1b shows OST substitutions significantly reduce the binding of a recombinant chimeric WT N2 protein more so than a stabilized recombinant chimeric N1 protein. Thus, it is possible the retention of binding is either subtype specific or due to the incorporation of the stabilizing mutations in N1. It is likely worth performing the same experiment with the same recombinant N1 with the WT sequence or at least clarifying this point in the conclusion, as it would provide insight into whether or not the OST mutations in N2 alter the substrate binding and hence the mimicry more so than N1.
2. To avoid the misconception that the prediction was based on two residues, it could be worth highlighting that the prediction combined the positioning of the DR motif in the CDR H3 region along with its extended length. For instance, a supplemental graph showing where the CDRH3 lengths they are searching for falls within the distribution of lengths in the NGS dataset.
3. The authors should clarify if the motif is DR or DR/RD as it is unclear in the current manuscript.
4. The last part of the last sentence of the abstract could be changed to something more directly related to the study such as "... antibodies that can potentially be elicited by an NA vaccine to provide broad protection against circulating influenza strains."
5. Like DA03E17, many of the pan NA antibodies isolated from H1N1 infected patients show much poorer binding to recent N2s and the large drop in affinity for KS17 indicates this is also true for DA03E17. It could be worth discussing alternative strategies for isolating N2 antibodies or proposing how the current isoforms would need to be altered for this class of NAs that are significantly resistant.
6. Line 231 – It may be more appropriate to conclude that the N245 glycan alone or in the context of the N146 glycan plays a prominent role in reducing the observed affinity reduction in recent N2s.
7. Lines 267-268 – Please clarify if the interactions with R100c are really "recapitulating" those of sialic acid and oseltamivir or forming different interactions with the same residues.

Robert Daniels, Ph.D.
PI Laboratory of Pediatric and Respiratory Viral Diseases
Division of Viral Products
Office of Vaccines Research and Review
Center for Biologics Evaluation and Research
U.S. Food and Drug Administration

Version 1:

Reviewer comments:

Reviewer #1

(Remarks to the Author)

The authors have responded thoughtfully and comprehensively to the reviewer's suggestions and most of the concerns have been addressed via additional experiments. In addition, the revisions have enhanced the interpretation of the results and improved the overall presentation structure of the work.

The current version of the manuscript represents a valuable contribution to the field.

(Remarks on code availability)

Reviewer #2

(Remarks to the Author)

I think Gyunghee Jo and co-authors have satisfactorily addressed all the comments from all 3 reviewers and have made significant improvements to the manuscript.

(Remarks on code availability)

Reviewer #3

(Remarks to the Author)

The revised version of the manuscript by Jo et al. is significantly improved from the original submission. The introduction provides more relevant background, and the manuscript contains new data and analyses that support conclusions which are more carefully written. Based on all these changes, I have no more concerns, and I think that this interesting study will be of broad interest to the field.

(Remarks on code availability)

Structural basis of broad protection against influenza virus by human antibodies targeting the neuraminidase active site via a recurring motif in CDR H3

Corresponding Authors: Julianna Han, Andrew B. Ward

Response to Reviewers

(Original reviewer's comments in **black**, our response in **blue**)

We sincerely thank all reviewers for their careful evaluation of our manuscript and for their insightful comments and questions. Addressing these points has helped us improve the clarity and scientific rigor of our study. In response to the reviewers' feedback, we performed new experiments including BLI using DA03E17 Fab, binding and inhibition assays with NAs carrying oseltamivir-resistant mutations, and determined four additional cryo-EM structures of DR motif antibodies. We also conducted sequence conservation analysis of NAs from animal-origin influenza A viruses and carried out more stringent bioinformatic searches. These additions allowed us to revise key sections and strengthen the mechanistic interpretations of our findings. Below, we provide detailed point-by-point responses to each comment.

Reviewer #1 (Remarks to the Author):

In this study Gyunghee and co-authors present the cryo-electron microscopy structures of the broadly IAV and IBV NA-inhibiting antibody DA03E17, which was previously isolated by the same group from an individual infected with A/H1N1pdm09 during the 2015–2016 influenza season. Similarly to other broadly NA-targeting monoclonal antibodies, DA03E17 binds to the highly conserved NA active site using its long CDR3 loop and mimicking the interaction of the sialic acid receptor. The authors reveal that an Asp–Arg (DR) motif, present in the CDRH3 of different broadly NA-inhibiting mAbs, represents the basis of this receptor mimicry and the broad activity against seasonal IAVs and IBVs. Moreover, this motif enables binding to oseltamivir-resistant and zoonotic IAV strains, including the currently circulating H5N1, thus highlighting its importance and in general the importance of this class of antibodies for universal influenza vaccine design.

Overall, the study presents novel structural information related to DA03E17 and other previously reported broadly NA-targeting mAbs and provides valuable insight on this broad activity and the possibility to elicit this class of antibodies through vaccination.

Major comments

Extended Figure 1b

1G05 was included as negative control since it binds to IBV NAs, but it would be important to have an idea of how DA03E17 activity against oseltamivir-resistant mutants compares with other previously described broadly NA inhibiting mAbs (e.g. 1G01 or FNI9). Considering that the authors presented structural data for 1G01 and FNI9, test these mAbs side by side should not be a problem.

We thank the reviewer for this helpful suggestion. To address this point, we included ELISA data for 1G01 in Supplementary Fig. 1b to enable comparison of its binding to oseltamivir-resistant NAs alongside DA03E17. We also added the following sentence to the Results section to describe this comparison:

Lines 134–138: “Compared to DA03E17, the previously described broadly neutralizing anti-NA mAb 1G0133 showed a greater difference in binding to oseltamivir-resistant BB18 N1 NAs and was particularly affected by the H274Y substitution, consistent with a previous report³⁶, while its binding to oseltamivir-resistant IN11 N2 NAs remained relatively consistent (Supplementary Fig. 1b).”

Although some of the oseltamivir-resistant mutations are associated with reduced neuraminidase activity, for those sNAp displaying enough sialidase activity the authors should produce ELLA results beside ELISA binding data.

We thank the reviewer for this helpful suggestion. In response, we performed ELLA using the same recombinant NA proteins tested by ELISA, including BB18 N1 variants carrying oseltamivir-resistant substitutions. The neuraminidase inhibition results for DA03E17 and 1G01 have been added to the manuscript (Supplementary Fig. 1c). Oseltamivir-resistant IN11 N2 mutants had to be excluded from ELLA because they did not exhibit detectable sialidase activity. We also added the following paragraph to the Results section to summarize these findings:

Lines 138–147: “We also assessed the NI activity of DA03E17 using ELLA with the same recombinant NA proteins containing oseltamivir-resistant mutations. DA03E17 retained inhibitory activity against BB18 N1 NAs carrying these mutations, with enhanced inhibition observed for H274Y, while activity was reduced against S246N and especially I222V variants. In contrast, 1G01 showed reduced inhibition against S246N and H274Y variants, consistent with its ELISA binding profile. Oseltamivir-resistant IN11 N2 mutants were excluded from ELLA due to lack of detectable sialidase activity, but DA03E17 exhibited similar levels of inhibition to 1G01 against the wild-type IN11 N2 (Supplementary Fig. 1c). These results demonstrate that DA03E17 retains binding and inhibitory activity against several oseltamivir-resistant NA variants, while also revealing mutation-specific differences in sensitivity.”

Extended Data Figure 3

Do the authors have any thought on the reason why DA03E17 have a significant lower affinity against N1 from H5N1 dairy cattle in comparison to the N1 from H1N1 CA09.

We thank the reviewer for raising this important point. In response to Reviewer #3’s suggestion to assess binding using the Fab form of DA03E17 to better isolate monovalent affinity from avidity effects, we performed additional BLI experiments using DA03E17 Fab (Supplementary Fig. 3). These experiments showed that DA03E17 Fab binds to the N1 from H5N1 A/Dairy Cattle/TX/24 (TX24 N1) with nanomolar affinity, which is comparable to its binding to CA09 N1.

In our earlier BLI experiments using IgG, we observed minimal dissociation with CA09 N1 and several other NAs, which resulted in apparent sub-picomolar K_D values. These values likely reflected avidity effects and limitations in kinetic resolution rather than a true difference in intrinsic binding affinity. The Fab-based measurements provide a more accurate assessment of monovalent affinity and do not indicate a substantial difference between TX24 N1 and CA09 N1.

To further investigate potential structural explanations, we predicted the structure of TX24 N1 using AlphaFold3, as no experimental structure is currently available. We then compared this

predicted model with the DA03E17-bound structure of CA09 N1. Most of the epitope residues were conserved between the two, suggesting that the overall binding interface is largely preserved. One notable difference was a substitution at position 347, where asparagine in CA09 N1 is replaced by tyrosine in TX24 N1 (N347Y). This change could potentially introduce a clash with the CDR H2 of DA03E17. However, despite this substitution, the Fab binding experiments did not reveal a significant loss in affinity, indicating that DA03E17 is still able to accommodate this variation without major disruption to binding.

Furthermore, the accompanying manuscript by Lederhofer and Borst et al., which was co-submitted and is under review alongside our study, demonstrates that DA03E17 retains neuraminidase inhibition activity against H5N1 virus. The observed activity was comparable to that against CA09 N1, supporting that DA03E17 effectively binds and inhibits TX24 N1.

Could the authors add the affinity of DA03E17 to the sNAP bearing oseltamivir-resistant mutations used for the ELISA assay reported in Extended Fig. 1. This would be helpful to have an idea of the affinity against the parental N1 or N2 and loss against ose-resistant mutations.

We thank the reviewer for this helpful suggestion. To address this point, we performed additional BLI experiments to measure the binding of DA03E17 to the oseltamivir-resistant BB18 N1 and IN11 N2 mutants used in the ELISA. Although DA03E17 binding to these mutants was observed by ELISA, binding was not detected by BLI under our experimental conditions, and thus affinities could not be determined. We speculate that the differences between the ELISA and BLI assay formats or the decreased binding affinity could contribute to the lack of detectable binding by BLI.

Figure 2

Any analysis to assess the conservation also across IAV NAs from animal circulating IAVs (mammals or avian species)?

We thank the reviewer for this suggestion. In response, we analyzed the conservation of DA03E17 epitope residues in N1 and N2 NAs from animal-origin HxN1 and HxN2 influenza A viruses that have circulated in avian and mammalian hosts between 1977 and 2023. This analysis is now included as Supplementary Fig. 7. We also added the following sentences in the Results section:

Lines 216–221: “We also assessed epitope conservation in animal-origin IAVs, analyzing N1 and N2 NA sequences from HxN1 and HxN2 viruses that have circulated in avian and mammalian hosts between 1977 and 2023 (Supplementary Fig. 7). DA03E17 epitope residues on N1 were generally well conserved in avian and mammalian HxN1 NAs, similar to human seasonal H1N1 viruses. In contrast, conservation in HxN2 NAs from animal-origin IAVs was lower compared to human H3N2 NAs.”

Despite the B-Yamagata-like lineage is considered extinct, it would be important to mention at least in the text if the authors observed any difference in conservation of the DA03E17 epitope between the NAs of Victoria-like and Yamagata-like lineages at least up to 2020?

We thank the reviewer for this suggestion. In response, we analyzed the conservation of the DA03E17 epitope in NAs from B/Yamagata/16/88-like IBVs circulating from 1988 to 2020. This analysis has been incorporated in Figure 2d (below the B/Victoria-lineage analysis), and we added the following sentence to the manuscript:

Lines 214–216: “Notably, although the B/Yamagata-like lineage has been considered extinct since 2020^{2,3}, the DA03E17 B/Vic NA epitope is also highly conserved in NAs of B/Yamagata/16/88-like IBVs (from 1988 to 2020; 25 out of 30, 83%).”

Furthermore, to compare the DA03E17 epitope conservation between B/Victoria-like and B/Yamagata-like IBVs, we revised the text as follows:

Lines 226–230: “Similarly, substitutions in the N2 epitope, from N147/R150/K344 (KS17 N2) to D147/H150/E344 (FJ02 N2), did not impair the NI activity of DA03E17, and substitutions in the influenza B NA epitope, from S244/R295/K343/E436 (CO17; B/Yamagata-lineage) to P244/S295/E343/T436 (B/Phuket/3073/2013; B/Victoria-lineage), had almost no effect on its NI activity³⁵. These results suggest that DA03E17 binding is resilient to these substitutions.”

Figure 2 panel d:

Based on lines 148-150 in the text: Shouldn't all the 19 NA residues contributing the most to the DA03E17 binding be denoted with symbols above the logo plots?

We appreciate the opportunity to clarify this point. In Figure 1f, we showed the interactions between the CDR H3 of DA03E17 and N1 NA, and for clarity, not all interacting epitopes were initially labeled. As these 19 N1 NA residues represent key epitopes that contribute most to DA03E17 binding, we have now labeled all of them in Figure 1f.

As described in the Methods, epitope residues were identified based on a buried surface area (BSA) greater than 5 Å². However, in the logo plot, only residues that form specific hydrogen bonds with DA03E17 were further highlighted with symbols (Figure 2d), as indicated in the figure key (Main chain H-bond / Side chain H-bond / Main and side chain H-bond). However, the original figure legend did not explicitly state that the symbols represent hydrogen-bonding epitopes, and we have now revised the legend to clarify this point:

“Symbols mark NA residues forming hydrogen bonds with DA03E17: circled bullets for main chain, open circles with rays for side chain, and circled bullets with rays for both.”

Along the same line Y406, I222, N294 are described as relevant catalytic site residues for DA03E7 CDRH3 binding to N1, but they are not indicated by any circle or circle with rays in the logoplot. If these amino acids are not so relevant, maybe it would be better to indicate in the text (148-150) only those residues that have crucial impact on DA03E17 binding.

We appreciate the reviewer’s comment. Y406, I222, and N294 are indeed important N1 epitope residues for DA03E17 binding. As noted above, symbols in the logo plot were used to indicate epitope residues that form specific hydrogen bonds with DA03E17. Y406, I222, and N294 contribute to binding primarily through hydrophobic and van der Waals interactions rather than hydrogen bonding, and therefore were not marked with symbols in the logo plot.

Related to Fig.5

The analysis of the DR motif prevalence in the human antibody repertoire appears quite limited and bit confused. Paragraph 296-309 is confusing as the authors move from the description of the deep sequencing dataset to state the structural characteristics of the previously described NA active site-targeting mAbs. Suggestion: move lines 296-298 describing the NGS approach and the number of human donors after the consideration on the broadly active mAbs and right before lines 307-309.

We have updated this section and we now include bioinformatic searches for precursors of the specific DR motif antibodies (DA03E17, 1G05, and Z2B3) in addition to the more general class of DR motif antibodies that were included in the first version. This analysis shows that 1G05 precursors (long CDR H3 with D gene in correct reading frame) with pre-existing DR motif at the correct position relative to the D gene should be targetable by vaccination and potentially also DA03E17 precursors. Z2B3 precursors with pre-existing DR motif were not found in the majority of donors and are therefore not a favorable vaccine target. This result has been added to Figure 5.

e

Search criteria:

i. Inferred D gene + Regular expression

DA03E17 (....GSG.....)

DA03E17+DR (....GSG..DR.....)

1G05 (....YSGYD.....)

1G05+DR (....YSGYDR.....)

Z2B3 (.....DT.MV.....)

Z2B3+DR (.....DT.MVDR.....)

ii. Flanked by any number of amino acids up to a maximum CDR H3 length of 30 amino acids

Additionally, we removed the description of the structural characteristics from this section because this was confusing as the reviewer pointed out and because the structural characteristics have already been described in previous sections of the paper.

Considering that 2 out of 4 mAbs presented in panel Fig.5a use IGHD5-18, it would be interesting to have an idea of the frequency of the BCR precursor also for the IGHD5 gene or at least to comment on that.

While we agree with the reviewers that this seems potentially interesting, in this case the two different alleles of D5-18 (D5-18*02 for 1G05 and D5-18*01 for Z2B3) encode very different amino acid sequences. Therefore, even though the D gene numbers are the same, the alleles and nucleotide/AA sequences are not interchangeable and not appropriate for making comparisons.

```
gtggatacagctatggttac D5-18*01
  G Y S Y G Y
gtgataatagtggtacgattac D5-18*02
  E Y S G Y D Y
gtggatatagtggtacgattac D5-12*01
  G Y S G Y D Y
```

Further, D5-18*02 was not found in our NGS dataset but D5-12 encodes a nearly identical sequence (1 nucleotide difference from D5-18*02) and because of this we searched using D5-12 for 1G05. We included this information in the methods section.

Discussion

Lines 334-337: "Considering its strong binding affinity to this bovine H5N1 NA..". This sentence appears an overstatement considering the affinity reported for DA03E17 in Extended Data Figure 3 against N1 from dairy cattle in comparison to N1 from Cal09.

We thank the reviewer for this helpful comment. We agree that the phrase "strong binding affinity" may overstate the difference, especially considering the IgG-based binding data shown in our earlier BLI experiments using IgG, which suggest a lower apparent affinity of DA03E17 for TX24 N1 compared to CA09 N1. However, our newly performed Fab BLI experiments (Supplementary Fig. 3) demonstrate that DA03E17 Fab binds TX24 N1 with nanomolar affinity, comparable to its affinity for CA09 N1, indicating that monovalent affinity is well preserved.

To address the reviewer's concern and to avoid potential overstatement, we have revised the sentence. The updated version reads as follows:

Lines 465–469: "Considering its nanomolar binding affinity to this bovine H5N1 NA, along with its in vivo protective efficacy against the typical avian H5N1 virus (A/Vietnam/1203/2004) isolated from a human, as demonstrated in our previous study³⁵, DA03E17 could also serve as a valuable therapeutic option if this virus were to spread more widely in humans."

This revised phrasing more accurately reflects the Fab binding data while maintaining the intended message regarding DA03E17's potential therapeutic value.

Lines 344-345: "DA03E17 and 1G01 may have the advantage in that they can accommodate this glycan without inducing structural changes in the NA loop, allowing for more stable and efficient binding". This statement is not supported by any data. Furthermore, based on Momont et. al 2023, in neutralization tests performed against recent H3N2 bearing N245 glycan FNI9 was consistently superior to 1G01 in its activity and 1G01 affinity was impacted to a higher extent than FNI9 by the N245 glycan. In addition, DA03E17 affinity is strongly reduced against N2 from H3N2 A/Kansas/2017 and Hong Kong /2019 in the Extended Data Figure 3. The authors should support this statement with in vitro tests comparing the activity of DA03E17 and 1G01 with FNI9 against N2 bearing N245 glycan or they should tune down this statement.

We thank the reviewer for this thoughtful comment. We agree that our original phrasing may have implied a functional advantage that was not directly supported by comparative experimental data. In response, we have revised the text to focus strictly on structural observations and removed any language implying superior binding efficiency. The revised paragraph now reads:

Lines 472–483: "Our cryo-EM structural analysis and MD simulations revealed that DA03E17 accommodates the N245 glycan of drifted H3N2 viruses by inducing conformational changes in the glycan. Similarly, 1G01 also induces conformational changes in the glycan itself, while FNI9 induces additional structural rearrangements in the NA loop to accommodate the N245 glycan. These differences suggest that DA03E17 and 1G01 engage the N245 glycan with minimal perturbation to the NA loop, representing an alternative mode of accommodation distinct from that of FNI9. The ability of these antibodies to structurally accommodate the N245 glycan of drifted H3N2 viruses may be of particular relevance, as H3N2-dominated influenza seasons are often associated with greater disease burden and higher hospitalization rates, especially in vulnerable populations⁶⁶. While these structural observations highlight varying modes of glycan engagement, further experimental data would be needed to determine whether these differences translate into functional advantages."

This revised text avoids any unverified claims of superiority and explicitly acknowledges the need for further experiments to assess functional consequences. We hope this addresses the reviewer's concern.

Minor comment

Fig.1h: labels for amino acids belonging to CDRL1, L3, and FR L3 are very difficult to read when printed on paper. Suggestion: use a darker border color.

We thank the reviewer for the suggestion. We have updated Figure 1h by changing the label font color to a darker shade for CDRL1, L3, and FR L3 to improve readability when printed.

Reviewer #2 (Remarks to the Author):

Jo et al. describe the molecular structure and binding epitope of the broadly protective human antibody DA03E17, which is directed against the enzymatic active site on NA, and elucidate the mechanism on how this mAb achieves broad reactivity against NA from A/H1N1, A/H3N2, and B/Victoria-lineage viruses. The same group had described this human mAb in a prior publication demonstrating the anti-viral effectiveness in mice against multiple Influenza strains (Yasuhara, A. et al. A broadly protective human monoclonal antibody targeting the sialidase activity of influenza A and B virus neuraminidases. *Nat Commun* 13, 6602 655 (2022).

In general, the manuscript is well written, and the data are of interest to the scientific community because it identifies the structural characteristics of this mAb, which enables its binding directly into the enzymatic pocket of NA, and how it accommodates the N-glycans of drifted N2 NAs. The authors show that this ability to accommodate the N-glycans of drifted N2 NAs seems to be a common feature with additional NA active site targeting mAb the authors evaluated. The authors also identified a motif that is conserved among several mAb NA active site targeting antibodies, indicating a common receptor mimicry strategy.

Comments the authors may want to consider:

- Trying to determine the prevalence of DA03E17-like mAb in the human antibody repertoire, the authors utilized an ultradeep next-generation sequencing (NGS) dataset from 14 healthy human donors for the precursor frequency search. By using the DR- or RD-motif, which seems to be common between six NA active site-targeting antibodies, and the characteristic long CDR H3 loop found in those mAb, the authors identified frequencies of such BCR sequences of 1 in 1,143 and 1 in 742, respectively, in all 14 donors. The authors acknowledge that the proportion of these DR or RD motif BCRs capable of maturing into functional NA antibodies remains to be determined, but it seems a little stretched to assume that just having a long CDR H3 loop with the DR or RD-motif is sufficient to allow the generation of these kinds of mAb. Therefore, the statement that the relatively high precursor BCR frequency the authors found with this method is “highlighting the prevalence of this motif and its potential as vaccine targeting” seems a big stretch in my opinion. Not only is it unlikely that the precursor BCR frequency identified by the authors really is representative of NA-specific mAb, but identifying strategies to expand those precursors and to induce large frequencies of NA active site-targeting antibodies through vaccination will be a significant challenge. This should be discussed more thoroughly on how the knowledge gained by the authors would allow the induction of DA03E17-like mAb. The authors may want to consider toning down this statement in the abstract.

We thank the reviewer for this thoughtful comment. To clarify, our analysis identified BCR sequences not only with a DR or RD motif within a long CDR H3, but specifically with the motif located centrally within the loop. Nevertheless, we agree that the presence of a DR or RD motif within a long CDR H3 is not sufficient to determine the full functional potential of a BCR precursor. To address this concern, we have revised the relevant sentence in the abstract to avoid overinterpretation. The updated sentence now reads:

Lines 035–037: “We also identified BCR sequences containing this DR motif across all donors in a healthy human repertoire database, suggesting that such precursors may be relatively common and have vaccine targeting potential.”

This revision reflects a more cautious interpretation of our repertoire analysis and avoids making claims about vaccine targeting potential based solely on motif prevalence. Additionally, as described in the following response and in the revised manuscript (Fig. 5), we performed further bioinformatic analysis to refine the precursor search criteria by incorporating the specific D gene and motif position relative to it. This additional analysis provides a more functionally relevant estimate of precursor frequencies for each antibody class.

- In addition to the DR- or RD-motive, would it be possible, based on the structural constraints of how those mAb can bind into the enzymatic pocket, to determine additional anchor amino acid requirements to refine the BCR-screening criteria and maybe determine a more realistic frequency of similar BCR, with potential of binding to NA, in the human repertoire?

We agree with the reviewer that this section was confusing and may have seemed misleading. While we do not observe any obvious anchoring residues adjacent to the DR/RD that are conserved among the different antibodies, the D gene will provide anchoring residues that will be unique for each antibody class (Ex: DA03E17, Z2B3 etc.) and we believe it will be beneficial to search for each antibody class independently. We have now carried out such bioinformatic searches for the three DR/RD antibodies for which the D gene could be inferred and included the new results in the manuscript in Figure 5. The new search criteria include the specific antibody D gene and reading frame and the DR motif at the same position relative to the D gene. This search identified 1G05 as the best vaccine target followed by DA03E17. Z2B3-like precursors with DR were not identified in the majority of donors.

- In order to provide more information on how frequent such NA active site-targeting mAb are after infection, performing competition ELISAs between the serum from those subjects used to identify the NA active site-targeting antibodies and the mAb DA03E17 would be informative. Performing direct NI inhibition in the MUNANA assay with the serum could also provide information about the contribution of NA active site-targeting antibodies in the serum to NI activity. I think it would be of high interest to the scientific community to understand how prevalent such Abs are in serum from

subjects after infection, not just as potential pre-cursor. That would provide a baseline which next generation vaccines (NA+HA) would have to reach to induce similar protection. Not too many of these NA active site-targeting antibodies have been identified in humans (so far, unless the accompanying manuscript by Lederhofer and Borst et al. describes additional, similar mAb against NA?; I don't have access to it), which would imply that the induction of such Ab even after infection is not very frequent. Knowing how frequent the mAb DA03E17 (or the other, similar mAbs) was in the person from which it was isolated would further increase the scientific value of this manuscript.

We agree that understanding the frequency of NA active site-targeting antibodies, whether from serum following infection or isolated monoclonal antibodies, is valuable to support precursor analysis. To address this, we used serological and structural techniques to assess active site antibodies in serum and characterize more DR motif containing monoclonal antibodies.

We conducted several experiments using serum samples from three human donors who were infected with A(H1N1)pdm09 virus during the 2015–2016 influenza season. These are the same donors from whom we previously isolated DA03E17 and other NA-targeting antibodies in our earlier study (Yasuhara et al., Nature Communications 2022). For each donor, we used serum samples collected at 1 and 3 months after symptom onset.

To assess the presence of active site-targeting antibodies in serum, we generated DA03E17 with a mouse IgG2a Fc domain and confirmed that it retained comparable binding affinity to NA as the original human IgG1 version. Using this mouse DA03E17, we performed competition ELISA against recombinant N1 NA from A/Michigan/45/2015 (Mich15), the strain circulating at the time of donor infection. While the serum samples showed detectable binding to Mich15 N1 NA, we did not observe any measurable competition with mouse DA03E17, suggesting that DA03E17-like antibodies were absent or below the detection threshold.

We next evaluated NA inhibition using the same serum samples in the NA-Star assay, a small molecule-based neuraminidase inhibition assay. Again, we did not detect inhibition of Mich15 N1 enzymatic activity by these sera.

Lastly, we purified polyclonal IgGs from the serum samples and performed electron microscopy-based polyclonal epitope mapping (EMPEM) using recombinant Mich15 N1. We observed some responses to the side face and corner regions of the NA tetramer, but no antibodies targeting the active site were detected.

Given that DA03E17 was originally isolated from one of these donors, we expected some level of active site-targeting response to be present in at least that donor's serum. However, none of the three experimental approaches revealed evidence of such antibodies. These results suggest that NA active site-targeting responses may be present at low levels after infection, may have decayed during long-term serum storage, or may fall below the detection limits of our assays.

While our serum-based data suggest that NA active site-targeting antibodies may be rare or difficult to detect, other lines of evidence indicate that these responses may be more common than they appear. In the accompanying manuscript by Lederhofer and Borst et al., which was co-submitted and is under review alongside our study, the authors also report two broadly cross-reactive antibodies (NCS.1.1 and NCS.1) targeting the NA active site via a central DR motif, isolated from an individual with a documented H3N2 infection. In addition, during the course of our revision, Wang et al. reported the isolation and structural characterization of a DR motif-containing antibody (4N2C402) that blocks the enzymatic activity of multiple NAs derived from diverse influenza strains (Wang et al., *Cell Host & Microbe*, 2025). The same group subsequently described four additional NA active site-targeting antibodies in a recent preprint (Lin et al., *bioRxiv*, 2025), all of which feature “xxxDRxxx” motifs in their CDR H3 regions highly similar to those described in our study, although structural data were not provided. Furthermore, Madsen et al. independently identified and structurally characterized a broadly reactive NA antibody (297) that uses the same DR motif to engage the active site via receptor mimicry (Madsen et al., *J. Exp. Med.*, 2025).

Building on these observations, we further identified five additional DR motif-bearing NA antibodies from patent and literature sources, and determined cryo-EM structures for four of them in complex with N1 NA (Fig. 6).

Fig. 6: Identification and structural characterization of additional DR motif NA antibodies.

These antibodies were derived from individuals with diverse exposure histories, including seasonal vaccination and H7N9 infection. Despite differences in germline gene usage and angles of approach, all four targeted the NA active site through a structurally conserved DR motif that mediated receptor mimicry. Their DR motifs engaged the same set of active site residues in a manner nearly identical to that seen in DA03E17 and other DR motif antibodies described in this study. This structural convergence across antibodies from unrelated donors, isolated under varied exposure conditions by different groups, highlights the recurring nature of this mode of recognition.

Together with the emerging studies noted above, our findings indicate that 12 structurally validated and broadly reactive antibodies targeting the NA active site through a conserved DR motif have been identified by multiple independent groups across distinct cohorts and viral exposures, supporting the potential of this antibody class as a target for broadly protective influenza vaccines. Although we were unable to detect active site-targeting antibodies in serum

in our experiments, we hope that the combination of our structural findings and the emerging evidence from other groups provides a meaningful response to the reviewer's question regarding their potential prevalence, and supports the notion that the NA active site is immunologically accessible and can be targeted through a convergent mechanism of receptor mimicry.

- In the prior publication in which the mAb DA03E17 was first described (Yasuhara, A. et al. 2022), an escape mutation in NA at position T439A was identified. In this new manuscript, T439 was no longer identified as an important anchor. The authors may want to consider discussing how the T439A substitution fits with their structural model on how the mAb DA03E17 binds to NA.

We thank the reviewer for the helpful comment. In response, we have added a description of the escape mutations observed at D151 and T439 in our earlier study. The escape mutations at D151 (D151G and D151N) can be explained by the direct contact between the CDR H3 of DA03E17 and D151 observed in our structure. As the reviewer pointed out, no direct contact between DA03E17 and T439 was observed. T439 is known to contribute to the stabilization of the 150-loop conformation through a hydrogen bond with T148 (Han et al., PLOS One, 2012; Wu et al., Scientific Reports, 2013), and therefore, the T439A substitution may affect DA03E17 binding indirectly by altering the conformation of the active site rather than through a direct interaction. However, since a structure of the T439A mutant NA is not available, we have refrained from making detailed speculations in the manuscript and have instead included the following statement:

Lines 187–192: “Substitutions at D151 (D151G and D151N), which were previously identified as major escape mutations for DA03E17 in our earlier study³⁵, can be explained by the direct contact between the CDR H3 of DA03E17 and D151 observed in our structure (Fig. 1f). In contrast, although a substitution at T439 (T439A) was also identified as an escape mutation, no direct contact between DA03E17 and T439 was observed in our structures. This substitution may affect binding indirectly by altering the conformation of the NA active site.”

- Line 213: the N146 glycan substitution is mentioned here for the first time in the manuscript. Could the authors mention when this N146 glycan substitution was first seen in circulating NA strains?

We thank the reviewer for this comment. To clarify, the N146 glycan is not a recent substitution but rather a glycan that has long been conserved in NAs of human and animal influenza viruses. To clarify this point, we have added the following sentence to the manuscript, also citing Östbye et al. (2020, Journal of Virology):

Lines 252–255: “Another N-linked glycan near the NA active site, the N146 glycan (Fig. 3a), is highly conserved in human and animal IAVs, including human H1N1

viruses circulating since 1918 and human H3N2 viruses circulating since 1968⁴³, and a corresponding N144 glycan is also conserved in IBVs.”

Minor corrections:

- Line 421: change “30 C” to “30°C”

We have corrected the typographical error by changing “30 C” to “30°C” on line 587 of the revised manuscript.

Reviewer #3 (Remarks to the Author):

The manuscript by Jo et al. builds upon three recently published studies (DOI: 10.1126/science.aay0678, DOI: 10.1038/s41586-023-06136-y, and DOI: 10.1038/s41467-022-34521-0) that focused on isolating and characterizing broadly inhibiting anti-NA antibodies from humans. This study combined new structural insight from the binding of the human antibody DA03E17 with the other available NA inhibitory antibody binding structures to identify a DR (Asp-Arg) motif in the CDR H3 that has the potential to mimic NA substrate interactions like those observed in DA03E17, resulting in the reported broad binding capacity against NAs. The prevalence of this motif was then screened in a healthy human donor BCR database suggesting it is commonly present in human and vaccines can potentially be developed to target these BCRs to elicit this broad class of anti-NA antibodies. Together, these results support the findings from previous studies that isolated this class of antibody against NA from different sources. It also speculates that vaccines could be developed to elicit these antibodies, supporting the concept that vaccines containing NA could provide increased breadth against circulating strains. Overall, this is a well written manuscript with clear data followed by the identification of a speculative motif that would be of broad interest with the minor revisions outlined below that include the need for more experimental or computational support for the putative DR motif and updating the introduction.

Main points

1. This study builds upon, and substantially benefits from, prior published data showing NA inhibitory antibodies mimic the binding of the sialic acid substrate. In its current form the introduction does not match the study and should be rewritten to focus more on NA and include the highly related and supportive published literature that this study is dependent on for identifying the putative DR/RD motif. This should strengthen rather than detract from the novelty of the study as it will increase the readability and support the more generalized statements about the mechanisms of action and binding of this class of antibodies. References should also be more explicit throughout the results section where appropriate.

We thank the reviewer for this thoughtful comment. We agree that the study benefits substantially from prior work demonstrating that NA active site-targeting antibodies can mimic the binding of the sialic acid substrate.

In response, we added two paragraphs further explaining the function of NA, resistance to sialic acid analogs, and previously described monoclonal antibodies (1G01, 1G05, and FN19) with characterized mechanisms of sialic acid mimicry. We also streamlined the Introduction by removing content that was not essential to this section. We retained the rest of the original text as it provides important background on influenza virus evolution, zoonotic risk, and vaccine design. We believe the integrated additions further strengthen the rationale for the study and improve the clarity and focus of the Introduction. In addition, we revised the Results section to make citations more explicit where relevant, including references to the N146 glycan and previously reported NA active site-targeting antibodies.

2. The DR/RD motif search for potential precursors in human antibody repertoires presented in figure 5 is currently very speculative and would benefit from more thorough analysis to put the data in context for the reader. For instance, some experimental data generating and testing the NA inhibitory nature of a few DR and RD motifs either by transfer to existing antibodies or as peptides could be performed to demonstrate if the motif is dependent or independent of the antibody context in which it resides given it appears to be a protruding domain that can potentially function independently. Alternatively, a more thorough bioinformatic analysis could be performed such as what is the prevalence of the DR motif in the subset of the BCR CDRH3 sequences with respect to a series of 2 random amino acid combinations or rank order the DR motif versus all other 2 amino acid combinations. What if any sequence conservation is present in the left and right sides of the sequences with the DR motif? Did any of the DR motifs in the database match any of the isolate antibodies? As it stands now, the reader has no way to know what the frequencies of 1 in 1,143 and 1 in 742 means and the conclusion is highly speculative.

We thank the reviewer for the insightful comments regarding the DR/RD precursor frequencies. We agree with the reviewer that the frequencies provided are speculative and we have included the following sentences in the results section: “Given the potential for additional requirements, only a subset of precursors meeting this bioinformatic definition would be expected to have potential to develop into NA active site-targeting antibodies. Therefore, this search represents an upper limit on the frequency of potential DR motif precursors that may be accessible to targeting by vaccination.” Importantly, we have carried out a more stringent bioinformatic search to complement the previous search by identifying precursors with not only the DR motif but also the specific features of the CDR H3 loops including the D gene identity and reading frame and the same D gene position relative to the DR motif. This search identified 1G05 as the best vaccine target followed by DA03E17, both have precursor frequencies within the range of other target antibodies that have shown promise in preclinical and clinical studies. We have cited those examples in the results. Z2B3-like precursors with DR were not identified in the majority of donors.

3. The KD calculations are all performed with IgG, which has a large avidity effect due to the two Fab arms. The authors should either repeat these measurements with FAb or be explicit about the IgG avidity in the text and use ‘apparent’ for all KD measurements.

We thank the reviewer for this important comment. In the revised manuscript, we now clarify that all KD values derived from IgG binding experiments represent apparent affinities, and we use the

term "apparent" consistently throughout the text. We also performed BLI experiments using monovalent DA03E17 Fab to assess binding independently of avidity. The Fab exhibited similar subtype-dependent binding patterns, with notably reduced binding to B/Victoria-lineage NA (CO17 B) compared to IgG. These results are described in the revised text and the following sentences, and are shown in Supplementary Fig. 3:

Lines 157–162: “DA03E17 Fab exhibited similar subtype-dependent binding patterns, with nanomolar to weaker nanomolar affinities for CA09 N1 sNAp, TX24 N1 sNAp, PT09 N2, and IN11 N2, and sub-micromolar affinities for KS17 N2, HK19 N2, and CO17 B. The binding of DA03E17 Fab to CO17 B NA was markedly reduced compared to IgG, indicating a greater contribution of avidity for this lineage (Supplementary Fig. 3). Together, these results confirm the broad cross-reactivity of DA03E17 to diverse influenza A and B virus NAs.”

Minor points

1. Extended data fig 1b shows OST substitutions significantly reduce the binding of a recombinant chimeric WT N2 protein more so than a stabilized recombinant chimeric N1 protein. Thus, it is possible the retention of binding is either subtype specific or due to the incorporation of the stabilizing mutations in N1. It is likely worth performing the same experiment with the same recombinant N1 with the WT sequence or at least clarifying this point in the conclusion, as it would

provide insight into whether or not the OST mutations in N2 alter the substrate binding and hence the mimicry more so than N1.

We thank the reviewer for this thoughtful comment. We understand the concern that the observed difference in OST substitution effects between N1 and N2 might be influenced by the presence of stabilizing mutations in the N1 sNAp construct used in our study. However, several lines of evidence suggest that the sNAp mutations are not expected to substantially impact the conformation of the NA active site or have a major influence on the interpretation of our binding data.

First, the sNAp mutations in N1 sNAp are located at the inter-protomeric interface and were introduced to stabilize the NA tetramer in its closed conformation. According to the original study reporting these mutations (Ellis et al., Nature Communications 2021), binding of active site-targeting antibody 1G01 to wild-type and sNAp-stabilized N1-MI15 was not substantially different in the presence of Ca^{2+} , indicating that the active site conformation remains largely intact.

Second, we directly compared our cryo-EM structures of the N1 sNAp and N2 NA with corresponding wild-type structures and observed no large conformational changes in the global fold, including the active site. As stated in the main text: “No large conformational changes in the global structure of the NA protein are observed compared with corresponding wild-type structures, as indicated by an RMSD of 0.356, 0.402, and 0.314 Å across all pairs for the N1, N2, and B NAs, respectively (Supplementary Fig. 5g–i).”

Third, in response to Reviewer #1, we included additional ELISA data for 1G01 in Supplementary Fig. 1b. Interestingly, 1G01 showed a greater difference in binding to wild-type versus oseltamivir-resistant N1 NAs, while its binding to oseltamivir-resistant N2 NAs remained relatively consistent. These data suggest that differential effects of OST substitutions are likely antibody-specific rather than purely subtype-specific. Furthermore, in the newly added ELLA data (Supplementary Fig. 1c), DA03E17 exhibited a range of NA inhibition (NI) activities against oseltamivir-resistant N1 NAs, which did not directly correlate with ELISA binding, further supporting the notion that OST substitution effects may reflect specific interactions between antibodies and altered NA active sites rather than the presence or absence of stabilizing mutations.

Taken together, these observations support the interpretation that the presence of sNAp mutations in N1 is unlikely to account for the observed differences in the effects of oseltamivir resistance-conferring mutations between N1 and N2.

2. To avoid the misconception that the prediction was based on two residues, it could be worth highlighting that the prediction combined the positioning of the DR motif in the CDR H3 region along with its extended length. For instance, a supplemental graph showing where the CDRH3 lengths they are searching for falls within the distribution of lengths in the NGS dataset.

We thank the reviewer for this suggestion and we have added a Supplementary Fig. 10 showing the CDR H3 length distribution for the entire NGS dataset and we have indicated the search area for the DR/RD antibodies. Additionally, we believe the more stringent bioinformatic search result avoids the misconception that the prediction was based on two residues.

Supplementary Fig. 10. CDR H3 length distribution in the NGS dataset for the 14 human donors. The search area for RD/DR antibodies is shaded gray.

3. The authors should clarify if the motif is DR or DR/RD as it is unclear in the current manuscript.

We thank the reviewer for the helpful comment. Among the receptor-mimicking antibodies compared in Figure 4, three contain a DR motif and one (FNI9) has an RD motif. Additionally, all of the newly identified antibodies shown in Figure 6 also carry a DR motif. As observed in the structure of FNI9, the DR and RD motifs differ only in directionality but adopt the same structural conformation and play similar functional roles. Therefore, we have unified the terminology as "DR motif" throughout the manuscript for clarity, reflecting both the structural equivalence and the predominance of this motif among the antibodies we analyzed.

4. The last part of the last sentence of the abstract could be changed to something more directly related to the study such as "... antibodies that can potentially be elicited by an NA vaccine to provide broad protection against circulating influenza strains."

We thank the reviewer for this helpful suggestion. We have revised the last part of the abstract to more directly reflect the potential implications of our findings, changing the sentence to:

Lines 038–039: "Our findings reveal shared molecular features in NA active site-targeting antibodies that can be harnessed to design broad, immune-focused influenza vaccines."

5. Like DA03E17, many of the pan NA antibodies isolated from H1N1 infected patients show much poorer binding to recent N2s and the large drop in affinity for KS17 indicates this is also true for DA03E17. It could be worth discussing alternative strategies for isolating N2 antibodies or proposing how the current isoforms would need to be altered for this class of NAs that are significantly resistant.

We thank the reviewer for this thoughtful comment. We agree that the reduced binding of DA03E17 to recent N2s, including A/Kansas/14/2017 (KS17), highlights an important challenge in targeting antigenically drifted N2 NAs. In response, we have revised the Discussion section to include this point and acknowledge that drifted N2s may pose a challenge for NA active site-targeting antibodies. We now also briefly mention potential future strategies, such as the use of N2-focused immunogens or structure-guided antibody engineering, to improve antibody recognition of recently evolved N2 variants. The revised text reads:

Lines 483–487: “Despite the broad cross-reactivity of DA03E17, we observed substantially reduced binding to recent N2s such as A/Kansas/14/2017, suggesting that drifted N2s may pose a challenge for NA active site-targeting antibodies. Future strategies could include immunogen designs that focus immune responses on conserved regions of N2 or structure-guided engineering of existing antibodies to improve their compatibility with recently evolved N2 NAs.”

6. Line 231 – It may be more appropriate to conclude that the N245 glycan alone or in the context of the N146 glycan plays a prominent role in reducing the observed affinity reduction in recent N2s.

Thank you for the suggestion. We have revised the sentence as follows to clarify that the N245 glycan, either alone or with the N146 glycan, plays a prominent role:

Lines 280–283: “Considering that the N146 glycan is also present in other N1 and N2 NAs, such as CA09 N1, PT09 N2, and IN11 N2, where DA03E17 Fab binds with a range of nanomolar affinity (Supplementary Fig. 3), the N245 glycan, either alone or in the context of the N146 glycan, plays a prominent role in the observed reduction in apparent affinity for recent N2s.”

7. Lines 267-268 – Please clarify if the interactions with R100c are really “recapitulating” those of sialic acid and oseltamivir or forming different interactions with the same residues.

Robert Daniels, Ph.D.
PI Laboratory of Pediatric and Respiratory Viral Diseases
Division of Viral Products
Office of Vaccines Research and Review
Center for Biologics Evaluation and Research
U.S. Food and Drug Administration

We thank the reviewer for this helpful suggestion. We have clarified the description to indicate that R100c forms contacts with the same NA residues (D151 and E227) as sialic acid and oseltamivir, but through different interactions, forming a salt bridge that strengthens binding:

Lines 319–322: “The side chain of R100c in the CDR H3 also forms contacts with NA residues D151 and E227. While sialic acid and oseltamivir also contact these residues, R100c in the CDRH3 contacts through different interactions, forming a salt bridge that strengthens binding.”

Structural basis of broad protection against influenza virus by human antibodies targeting the neuraminidase active site via a recurring motif in CDR H3

Corresponding Authors: Julianna Han, Andrew B. Ward

Response to Reviewers – 2nd revision

(Original reviewer's comments in **black**, our response in **blue**)

We thank all three reviewers for their thoughtful and constructive feedback. We appreciate the time and effort they took to evaluate our manuscript and are glad that the revised version was well received.

Reviewer #1 (Remarks to the Author):

The authors have responded thoughtfully and comprehensively to the reviewer's suggestions and most of the concerns have been addressed via additional experiments. In addition, the revisions have enhanced the interpretation of the results and improved the overall presentation structure of the work. The current version of the manuscript represents a valuable contribution to the field.

Thank you for the thoughtful and encouraging comments. We're glad the revisions helped clarify and improve the manuscript.

Reviewer #2 (Remarks to the Author):

I think Gyunghee Jo and co-authors have satisfactorily addressed all the comments from all 3 reviewers and have made significant improvements to the manuscript.

Thank you for your kind feedback. We're happy the revisions addressed the concerns satisfactorily.

Reviewer #3 (Remarks to the Author):

The revised version of the manuscript by Jo et al. is significantly improved from the original submission. The introduction provides more relevant background, and the manuscript contains new data and analyses that support conclusions which are more carefully written. Based on all these changes, I have no more concerns, and I think that this interesting study will be of broad interest to the field.

Thank you for the supportive review. We appreciate that the new data and revised framing were well received.